# *Shigella flexneri* evades LPS ubiquitylation through IpaH1.4-mediated degradation of RNF213

Katerina Naydenova[1,7], Keith B. Boyle[1,7], Claudio Pathe[1,4,7], Prathyush Pothukuchi[1], Ana Crespillo-Casado[1,5], Felix Scharte[1], Pierre-Mehdi Hammoudi[1,6], Elsje G. Otten[1], Neal M. Alto[2] & Felix Randow[1,3]✉

Pathogens have evolved diverse strategies to counteract host immunity. Ubiquitylation of lipopolysaccharide (LPS) on cytosol-invading bacteria by the E3 ligase RNF213 creates 'eat me' signals for antibacterial autophagy, but whether and how cytosol-adapted bacteria avoid LPS ubiquitylation remains poorly understood. Here, we show that the enterobacterium *Shigella flexneri* actively antagonizes LPS ubiquitylation through IpaH1.4, a secreted effector protein with ubiquitin E3 ligase activity. IpaH1.4 binds to RNF213, ubiquitylates it and targets it for proteasomal degradation, thus counteracting host-protective LPS ubiquitylation. To understand how IpaH1.4 recognizes RNF213, we determined the cryogenic electron microscopy structure of the IpaH1.4–RNF213 complex. The specificity of the interaction is achieved through the leucine-rich repeat of IpaH1.4, which binds the RING domain of RNF213 by hijacking the conserved RING interface required for binding to ubiquitin-charged E2 enzymes. IpaH1.4 also targets other E3 ligases involved in inflammation and immunity through binding to the E2-interacting face of their RING domains, including the E3 ligase LUBAC that is required for the synthesis of M1-linked ubiquitin chains on cytosol-invading bacteria downstream of RNF213. We conclude that IpaH1.4 has evolved to antagonize multiple antibacterial and proinflammatory host E3 ligases.

The intracellular lifestyle has evolved multiple times amongst pathogenic bacteria, enabling them to gain access to cellular nutrients and avoid potent extracellular immunity[1]. However, most intracellular bacteria reside in membrane-delimited compartments, with only a select few succeeding in colonizing the host cytosol[2]. Access to cellular nutrients is arguably easier in the cytosol, suggesting that cytosolic antibacterial immunity is particularly potent as it successfully prevents colonization by most facultatively intracellular bacteria. To sense and curb bacterial infection in the cytosol, a variety of cell-autonomous immune mechanisms exist. These include inflammasome activation, the induction of cell death, and antibacterial autophagy. The lattermost relies on pairs of 'eat me' signals associated with bacteria and cognate receptors that promote autophagy[3], for example, sphingomyelin on stressed but otherwise intact bacteria-containing vacuoles sensed by TECPR1 (ref. 4), glycans on broken bacteria-containing vacuoles sensed by galectin 8 (ref. 5) and ubiquitin conjugated to the bacterial surface

[1]Division of Protein and Nucleic Acid Chemistry, MRC Laboratory of Molecular Biology, Cambridge, UK. [2]Department of Microbiology, University of Texas (UT) Southwestern Medical Center, Dallas, TX, USA. [3]Department of Medicine, Addenbrooke's Hospital, University of Cambridge, Cambridge, UK. [4]Present address: AstraZeneca, The Discovery Centre, Cambridge Biomedical Campus, Cambridge, UK. [5]Present address: Origin Sciences, Cambridge, UK. [6]Present address: MabDesign, Lyon, France. [7]These authors contributed equally: Katerina Naydenova, Keith B. Boyle, Claudio Pathe. ✉e-mail: randow@mrc-lmb.cam.ac.uk

and sensed by multiple cargo receptors including NDP52, TAX1BP1, Optineurin and SQSTM1 (refs. [6–10]).

We recently discovered that the initial step of ubiquitin coat formation on cytosol-exposed *Salmonella enterica* serovar Typhimurium (*Salmonella* Typhimurium) is catalyzed by the E3 ligase RNF213, which ubiquitylates bacterial lipopolysaccharide (LPS) in the first example of ubiquitylation targeting a nonproteinaceous substrate[11]. Ubiquitylation of LPS results in the downstream recruitment of LUBAC, a multimeric E3 ligase that adds M1-linked ubiquitin chains to RNF213-initiated ubiquitin coats[12,13]. In addition to Gram-negative bacteria, RNF213 also restricts pathogens that do not produce LPS, including Gram-positive bacteria, the apicomplexan parasite *Toxoplasma gondii* and certain viruses[14–19]. Mutations in *RNF213* cause Moyamoya disease, a rare cerebrovascular disorder caused by intimal thickening and occlusion of the terminal portion of the internal carotid artery[20–22].

Given that certain bacteria successfully colonize the host cytosol, we wondered whether and how they escape LPS ubiquitylation. *Shigella flexneri* is an example of a cytosol-dwelling Gram-negative bacterium that has evolved a plethora of mechanisms to avoid clearance by cell-autonomous immunity, including antibacterial autophagy, to establish its replicative niche in the cytosol[23–25]. These adaptations include the IpaH effector proteins, a family of bacterial E3 ubiquitin ligases that are secreted into the cytosol of infected cells, where they engage the host ubiquitin–proteasome system[26–29]. IpaH proteins comprise a short unstructured N terminus, a leucine-rich repeat (LRR) mediating substrate recognition, and a C-terminal so-called 'novel E3 ligase' (NEL) domain. It has been suggested that, in the absence of substrate, IpaH ligases are autoinhibited by their LRR domain through blockade of the catalytic cysteine in the E3 domain[30–33]. *S. flexneri* encodes up to 12 functional IpaH proteins, five of which are found on the *Shigella* virulence plasmid: *ipaH1.4*, *ipaH2.5*, *ipaH4.5*, *ipaH7.8* and *ipaH9.8*. Examples of IpaH ligases and their immunoregulatory host substrates include IpaH9.8 and guanylate-binding proteins (GBPs)[34–36], IpaH7.8 and gasdermins B and D[37–39], IpaH9.8 and NEMO[40] and IpaH1.4 and LUBAC[41,42]. In all known cases, the IpaH ligase targets important antibacterial host proteins for proteasomal degradation through conjugation of polyubiquitin chains. IpaH proteins, thus, synergistically enable *Shigella* to retain actin-dependent motility, block pyroptotic cell death and reduce NF-κB activation, together dampening the antibacterial and proinflammatory response of their host cells.

In this study, we investigate whether *S. flexneri* antagonizes LPS ubiquitylation in the cytosol of infected cells. We find that *S. flexneri* indeed does not succumb to LPS ubiquitylation, because RNF213, the E3 ligase catalyzing LPS ubiquitylation[11], is bound, ubiquitylated and targeted for proteasomal degradation by the bacterial effector protein IpaH1.4. Cryogenic electron microscopy (cryo-EM) revealed that IpaH1.4 binds to the RING domain of RNF213. Remarkably, *S. flexneri* deploys the same IpaH1.4 effector to also target a RING domain in LUBAC[41,42] (known to produce M1-linked ubiquitin chains on bacteria downstream of RNF213)[12] and in several other RING-containing proteins, thus revealing how a single bacterial effector antagonizes multiple steps in a complex cascade of antibacterial host proteins.

## Results

### *S. flexneri* produces a *trans*-acting factor to avoid LPS ubiquitylation

We previously demonstrated that the facultatively cytosol-dwelling bacterium *Salmonella* Typhimurium undergoes RNF213-dependent LPS ubiquitylation[11]. To test whether a professionally cytosol-invading bacterium is similarly recognized, we infected HeLa cells, a human epithelial cell line, with *S. flexneri*. We noticed that, unlike *Salmonella* Typhimurium, *S. flexneri* did not become ubiquitin-coated in the host cytosol (Fig. 1a). To directly address LPS ubiquitylation, we immunoblotted lysates of bacteria isolated from cells at 4 h after infection with an antibody specific to conjugated ubiquitin (FK2) (Fig. 1b). Immunoblotting

revealed an LPS–ubiquitin smear and a characteristic banding pattern in the positive control samples of wild-type (WT) and Δ*rfaL Salmonella* Typhimurium, respectively, which are caused by the ubiquitylation of high-molecular-weight and low-molecular-weight LPS in the two strains[11]. In contrast, ubiquitylated LPS was not detected in lysates from cells infected with either WT or Δ*rfaL S. flexneri*. In coinfection experiments, *Salmonella* was protected from ubiquitylation when present together with *Shigella* in the same host cell (Fig. 1c). This observation suggests the existence of a secreted *Shigella* effector acting in *trans* to antagonize LPS ubiquitylation on *Salmonella*. We also noticed that lack of ubiquitylation of *S. flexneri* correlated with lack of recruitment of endogenous RNF213 to the bacteria (Fig. 1a,c) and recruitment of RNF213 to the surface of cytosolic *Salmonella* was abolished in cells coinfected with *Shigella*. We, therefore, conclude that the *trans*-acting factor produced by *Shigella* directly antagonizes RNF213 and is not merely removing ubiquitin from bacteria.

### Both IpaH1.4 and IpaH2.5 (IpaH1.4/2.5) bind and ubiquitylate RNF213

To identify the *trans*-acting factor antagonizing RNF213, we tested whether any of the secreted IpaH-family effectors from *S. flexneri* can bind to RNF213 in a pulldown assay using purified recombinant proteins. Among all 12 IpaH effectors, only the virulence plasmid-encoded IpaH1.4/2.5 proteins bound human RNF213 (Fig. 2a). Cross-reactivity of IpaH1.4/2.5 toward RNF213 is not unexpected as they are closely related proteins with identical LRRs, the canonical substrate-binding site in IpaH proteins (Extended Data Fig. 1a). To study the effect of IpaH1.4/2.5 on RNF213, we generated HeLa cells expressing GFP-tagged IpaH proteins or their catalytically inactive variants, in which the catalytically active cysteine in the NEL domain is replaced with an alanine to abolish E3 ubiquitin ligase activity. Expression of WT IpaH1.4/2.5, but not their catalytically inactive variants (IpaH1.4/2.5-C368A) or IpaH7.8, depleted RNF213 in the host cells (Fig. 2b). Concomitantly, all active IpaH ligases but not their catalytically impaired variants induced their own degradation when overexpressed in mammalian cells (Fig. 2b). To test the effect of IpaH1.4/2.5 on LPS ubiquitylation, we infected IpaH-expressing cells with *Salmonella* Typhimurium Δ*rfaL*, which is susceptible to LPS ubiquitylation and whose truncated LPS produces a characteristic banding pattern in the immunoblot for conjugated ubiquitin in heat-cleared bacterial lysate from infected cells[11] (Fig. 2b). Expression of IpaH7.8, which neither binds to nor depletes RNF213, had no effect on the ubiquitylation of LPS, while, in contrast, WT IpaH1.4/2.5 prevented LPS ubiquitylation, consistent with their ability to deplete RNF213. Catalytically inactive IpaH1.4/2.5-C368A did not prevent LPS ubiquitylation and did not reduce RNF213 protein levels, suggesting that mere binding of IpaH1.4/2.5 to RNF213 is insufficient to inhibit RNF213 activity. We conclude that IpaH1.4/2.5 antagonize LPS ubiquitylation through depletion of cellular RNF213 protein levels in a manner that requires IpaH E3 ubiquitin ligase activity.

To investigate the mechanism of RNF213 depletion by IpaH1.4/2.5, we tested whether these proteins can ubiquitylate RNF213 in vitro. To avoid any unwanted background stemming from autoubiquitylation of RNF213, we used RNF213-H4509A, a catalytically inactive variant[11], as substrate in the reaction (Extended Data Fig. 2a). We found that in the presence of recombinant ubiquitin and E1 and E2 enzymes, RNF213 is ubiquitylated by IpaH1.4/2.5 but not by their catalytically inactive variants (IpaH1.4/2.5-C368A) or IpaH9.8 (Fig. 2c and Extended Data Fig. 2b). In summary, we discovered that IpaH1.4/2.5 can bind and ubiquitylate RNF213 in vitro and, when recombinantly expressed in cells, target RNF213 for degradation, which prevents LPS ubiquitylation upon infection with *Salmonella* Typhimurium.

### IpaH1.4 protects *S. flexneri* from LPS ubiquitylation

We next tested whether the lack of RNF213-mediated LPS ubiquitylation in cells infected with *S. flexneri* is because of the action of IpaH1.4/2.5 on

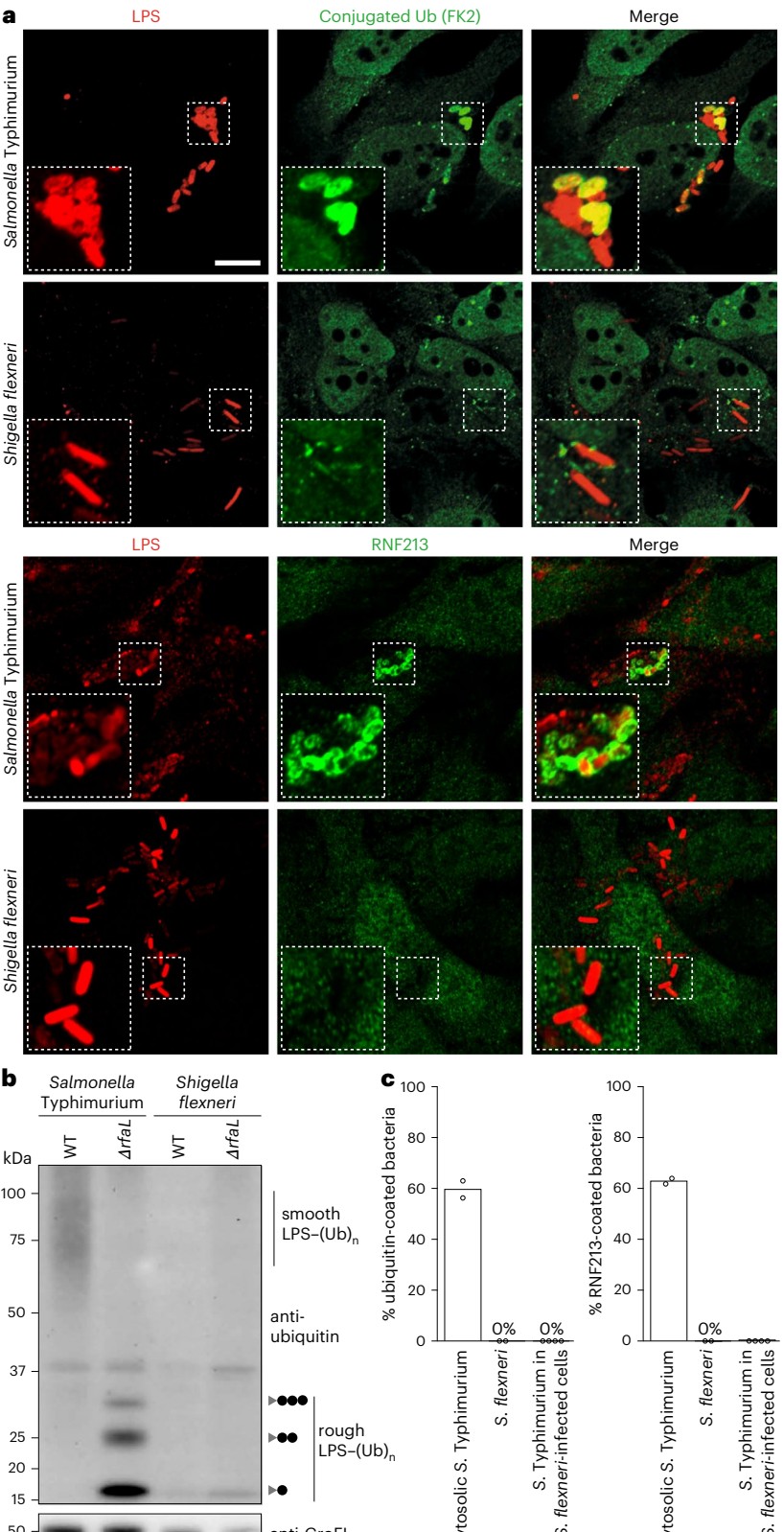

**Fig. 1 | A *Shigella*-derived *trans*-acting factor antagonizes LPS ubiquitylation by RNF213. a**, Representative confocal micrographs of HeLa cells at 5 h after infection with *Salmonella* Typhimurium (top) or *S. flexneri* (bottom), immunostained for LPS (red) and conjugated ubiquitin (FK2) (green) or LPS (red) and RNF213 (green) as indicated. Dashed squares, representative areas magnified ×2.5 in the insets. Scale bar, 10 μm. The micrographs are representative of *n* = 3 experiments. **b**, Immunoblot analysis of the indicated strains of *Salmonella* Typhimurium and *S. flexneri* extracted from HeLa cells at 5 h after infection.

Gray triangle, *ΔrfaL* LPS; black circle, ubiquitin; GroEL, loading control for bacterial lysates. The LPS fractions were isolated by heat clearance of bacterial lysates and probed for conjugated ubiquitin with FK2 antibody. The loading control (GroEL) was probed for in non-heat-cleared bacterial lysates. The results are representative of *n* = 3 experiments. **c**, Percentage of bacteria positive for conjugated ubiquitin (detected by FK2 antibody) (left) and RNF213 (right) at 4 h after infection in HeLa cells. Data are averaged from *n* = 2 experiments for singly infected cells or *n* = 4 experiments for the double infection.

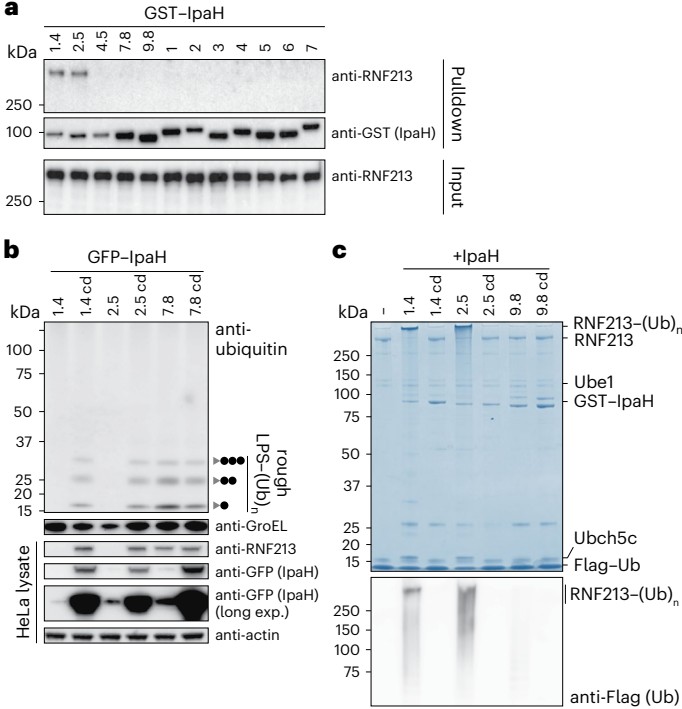

**Fig. 2 | IpaH1.4/2.5 bind to, ubiquitylate and antagonize RNF213. a**, HeLa cell lysate incubated with beads displaying the indicated GST-tagged IpaH proteins. Bound RNF213 was detected by western blot. The results are representative of *n* = 3 experiments. **b**, Immunoblot analysis of *Salmonella* Typhimurium *ΔrfaL*, extracted from HeLa cells expressing the indicated GFP-tagged IpaH proteins. Gray triangle, *ΔrfaL* LPS; black circle, ubiquitin; GroEL, loading control for bacterial lysates; cd, catalytically dead (E3 inactive) variants; actin, loading control for HeLa lysates. The LPS fractions were isolated by heat clearance of bacterial lysates and probed for conjugated ubiquitin with FK2 antibody. The loading control (GroEL) was probed for in non-heat-cleared bacterial lysates. All other proteins were probed for in host cell lysates. The results are representative of *n* = 4 experiments. **c**, Coomassie-stained gel of an in vitro ubiquitylation reaction containing Flag-tagged ubiquitin, UBE1, UBCH5C, the indicated IpaH proteins and enzymatically inactive RNF213-H4509A (Extended Data Fig. 2a) as substrate. The majority of RNF213 is ubiquitylated by IpaH1.4/2.5, yielding a product of substantially increased molecular mass that migrates more slowly in the gel, as indicated. Bottom, immunoblot for Flag-tagged ubiquitin. The results are representative of *n* = 3 experiments.

RNF213. We infected cells expressing GFP–RNF213 with Ruby-labeled *S. flexneri* WT, *ΔipaH1.4* or *ΔipaH2.5* strains for analysis by flow cytometry (Fig. 3a and Extended Data Fig. 3). When infected with WT bacteria or *S. flexneri ΔipaH2.5*, around 30% of the infected cells lost their GFP fluorescence at 5 h after infection, indicative of RNF213 degradation. In contrast, upon infection with *S. flexneri ΔipaH1.4* no measurable RNF213 degradation occurred. Complementation of *S. flexneri ΔipaH1.4* with plasmid-encoded *ipaH1.4* under control of an inducible promoter fully recovered the ability to degrade RNF213 in infected cells (Extended Data Fig. 4a). Endogenous RNF213 became similarly depleted in HeLa cells infected with *S. flexneri* WT or *ΔipaH2.5* but not *ΔipaH1.4*, as assessed by immunostaining of endogenous RNF213 in infected cells or by western blotting of lysates of infected cells purified by fluorescence-activated cell sorting (Extended Data Fig. 4b,c). We conclude that IpaH1.4 but not IpaH2.5 is required for RNF213 degradation by *S. flexneri*. To obtain further insight into the differential contribution of IpaH1.4 and IpaH2.5 to RNF213 degradation, we analyzed RNA extracted from cells infected with WT *S. flexneri* and confirmed that transcripts for both *ipaH1.4* and *ipaH2.5* were present, indicating that both genes are expressed and that likely a downstream mechanism (for example, lack of IpaH2.5 protein secretion) limits the ability of IpaH2.5

to target RNF213 (Extended Data Fig. 5a). To investigate how ubiquitylated RNF213 is degraded, we infected GFP–RNF213 reporter cells with *S. flexneri* and treated them with either carfilzomib or bafilomycin A1, inhibitors of the proteasome or autophagy pathway, respectively (Fig. 3a and Extended Data Fig. 3c). Only treatment with carfilzomib prevented degradation of RNF213 upon infection with *Shigella*, indicating that IpaH1.4-induced degradation of RNF213 is mediated by the proteasome, as is the case for other IpaH–host effector pairs[35,37,40,41].

We next tested whether the modulation of RNF213 protein levels by IpaH1.4 has functional consequences. We observed that knockout (KO) of *ipaH1.4* allowed for recruitment of RNF213 to *S. flexneri* and rendered the bacteria susceptible to RNF213-mediated ubiquitylation (Fig. 3b–d and Extended Data Fig. 5b,c). In contrast, *S. flexneri ΔipaH2.5* was indistinguishable from the WT strain in this experiment, confirming that IpaH1.4 is the sole IpaH-family effector antagonizing RNF213 under these conditions. Knockout of *mxiE*, a transcriptional regulator that stimulates expression of secreted effectors[43], including the IpaH family, also rendered *S. flexneri* susceptible to coating with ubiquitin and RNF213 (Fig. 3c). Next, we specifically tested the effect of IpaH1.4 on LPS ubiquitylation (Fig. 3d and Extended Data Fig. 5c). We found that deletion of *ipaH1.4* but not *ipaH2.5* or *ipaH9.8* rendered *S. flexneri* LPS susceptible to ubiquitylation. We conclude that IpaH1.4 causes proteasome-mediated degradation of RNF213 and thereby prevents LPS ubiquitylation, whereas IpaH2.5 does not contribute to the phenotype, at least in the context of *S. flexneri* infection in cell culture. Our observation is consistent with previous findings regarding IpaH1.4/2.5 and LUBAC, where both effectors targeted LUBAC in vitro but only IpaH1.4 affected LUBAC in infected cells[41].

We then investigated whether RNF213-mediated LPS ubiquitylation restricts the proliferation or the cell-to-cell spread of *S. flexneri* lacking IpaH1.4. We found that *S. flexneri ΔipaH1.4* proliferated (Fig. 3e) and spread (Fig. 3f) in a manner comparable to WT bacteria. This result indicates that RNF213-mediated ubiquitylation of bacterial LPS is insufficient to restrict *Shigella* proliferation or cell-to-cell spread in a cell-based infection model, a perhaps surprising finding as RNF213 is crucial for restricting the proliferation of *Salmonella* Typhimurium in the host cytosol[11]. It is, therefore, likely that *Shigella* has evolved additional strategies to block the downstream effects of RNF213-mediated ubiquitylation. These might include factors that block the recruitment of autophagy receptors and inhibit NF-κB activation (for example, IcsB, IpaH9.8, OspI and OspZ)[24,40,41,44–46] or completely orthogonal means, such as actin-mediated motility, which may allow escape from the consequences of LPS ubiquitylation.

## IpaH1.4 binds the RING domain of RNF213

Lastly, to gain molecular insights into the interaction between IpaH1.4/2.5 and RNF213, we determined the structures of human RNF213 bound to IpaH1.4/2.5 using cryo-EM (Fig. 4a–f, Table 1 and Extended Data Figs. 6 and 7). The cryo-EM maps show that the LRR domains of IpaH1.4/2.5 interact with the RING domain of RNF213 (Fig. 4b–d and Extended Data Fig. 6), whereas the unstructured N termini and the catalytic NEL domains of IpaH1.4/2.5 remain unresolved. Focused refinement of the RNF213–IpaH1.4/2.5 structure around the RING domain of RNF213 reveals molecular details of their interaction to 3.3-Å resolution (Fig. 4d), a higher-resolution reconstruction of the RNF213 RING domain than that previously achieved in published apo structures of RNF213 (refs. 14,47). IpaH1.4/2.5 interact exclusively with the RING domain of RNF213. Unsurprisingly, given the sequence identity of their LRR domains, IpaH1.4/2.5 engage the RING domain of RNF213 in exactly the same way (Extended Data Fig. 6). Binding is mediated by the canonical concave substrate-binding side of the LRR, involving an extensive interaction interface spanning from strand β2 to β7. Several hydrophobic interactions (for example, IpaH1.4/2.5 F120 with RNF213 F3992 and IpaH1.4/2.5 V177 with RNF213 L4036) and hydrogen-bonded amino acid pairs (for example, IpaH1.4/2.5 K100 with RNF213 D4013,

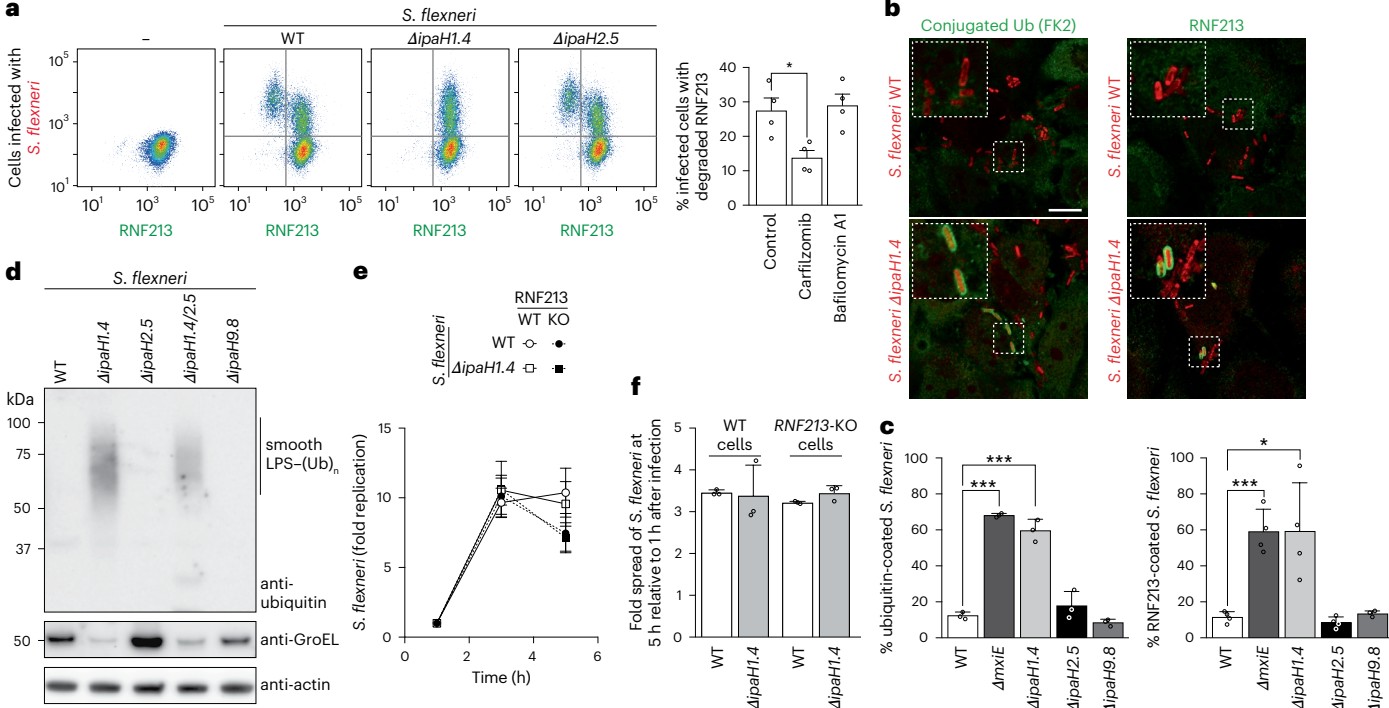

**Fig. 3 | IpaH1.4-induced proteasomal degradation of RNF213 protects *Shigella* from LPS ubiquitylation. a**, Flow cytometry of GFP–RNF213-expressing MEFs, infected with the indicated strains of Ruby-labeled *S. flexneri* for 5 h. Quadrants depicted were used to quantify the percentage of infected cells in which degradation of RNF213 is observed. Note that cells with a higher bacterial burden are more likely to have lost RNF213 (Extended Data Fig. 3a,b). Bar graph shows the fraction of infected cells where RNF213 is degraded (top left quadrant in Extended Data Fig. 3c) in the presence of the indicated inhibitors. Data are from $n = 4$ experiments performed in duplicate. Bars denote the mean and error bars denote the s.d. *$P = 0.018$ (two-sided unpaired *t*-test). **b**, Representative confocal micrographs of HeLa cells at 4 h after infection with the indicated strains of Ruby-labeled *S. flexneri*, immunostained for conjugated ubiquitin (FK2) (left) or RNF213 (right). Dashed squares are magnified ×2.5 in the insets. Scale bar, 10 µm. The corresponding micrographs of individual color channels are shown in Extended Data Fig. 5b. The micrographs are representative of $n = 4$ experiments. **c**, Percentage of the indicated *S. flexneri* strains positive for conjugated ubiquitin (detected by FK2 antibody) (left) and GFP–RNF213 (right) at 4 h after infection in MEFs expressing human GFP–RNF213. Data are from $n = 3$ experiments, with bars denoting the mean and error bars denoting the s.d. ***$P = 1.7 × 10^{-6}$, $P = 2.4 × 10^{-4}$ and $P = 3.1 × 10^{-4}$; *$P = 0.012$ (two-sided unpaired *t*-tests). **d**, Immunoblot analysis of the indicated strains of *S. flexneri*, which were extracted from MEFs at 4 h after infection. The LPS fractions were isolated by heat clearance of bacterial lysates and probed for conjugated ubiquitin with FK2 antibody. The loading control (GroEL) was probed for in the non-heat-cleared bacterial lysates. The host cell loading control (actin) was probed for in clarified host cell lysates (Extended Data Fig. 5c). The results are representative of $n = 2$ experiments. **e**, Fold replication of the indicated strains of *S. flexneri* in WT and *RNF213*-KO HeLa cells. Bacteria were counted on the basis of their ability to grow on TSB plates. Proliferation was quantified from a single experiment, representative of two, including $n = 6$ technical replicates. Data are expressed as the mean ± s.d. **f**, Fold cell-to-cell spread of the indicated strains of *S. flexneri* in WT and *RNF213*-KO HeLa cells at 5 h after infection relative to 1 h after infection, quantified from $n = 3$ experiments. Data are expressed as the mean ± s.e.m.

IpaH1.4/2.5 R215 with RNF213 Q4029 and IpaH1.4/2.5 R157 with RNF213 C4035 carbonyl O) stabilize the interface (Fig. 4e and Extended Data Fig. 8). Overall, the RNF213 RING domain docks into a net positively charged cavity in the LRR (Fig. 4f) and into two hydrophobic patches formed by strands β6 and β8–β10.

Substitutions of key interfacial residues, including L4036E in RNF213 and K100A, F120E, R215A, V217E and F238D in IpaH1.4 abolished or reduced binding between RNF213 and IpaH1.4 (Extended Data Fig. 8). To test whether the binding interface identified is important for the degradation of RNF213 by IpaH1.4 during infection, we generated *S. flexneri* expressing HA-tagged IpaH1.4 alleles and tested their ability to degrade RNF213. While WT IpaH1.4 effectively degraded RNF213, neither IpaH1.4-F238D nor IpaH1.4-R215A;V217E did so, similarly to catalytically inactive IpaH1.4-C368A (Extended Data Fig. 9).

Focused refinements of the other domains of RNF213 revealed that the binding of IpaH1.4/2.5 does not cause any conformational changes in comparison to the previously published apo structure of human RNF213 (ref. 14). The flexible RNF213 loop between Q4099 and E4112, which is not resolved in the RNF213 apo structures[14,47] or in the IpaH1.4/2.5-bound structures reported here, must point toward the convex LRR face without forming stable interactions. Although we

did not resolve the NEL domain of IpaH1.4/2.5, locating the LRR at the RNF213 RING domain allows us to predict the region of RNF213 accessible to the ubiquitylating activity of IpaH1.4/2.5 (Fig. 4b). Accounting for the length of the flexible linker between LRR and NEL domains in IpaH1.4/2.5, we predict that the NEL catalytic domain will have access to the CBM20 carbohydrate-binding module of RNF213, to its E3 module (particularly the M-lobe of the E3 shell) and possibly to 32 lysine residues in the unstructured N terminus of RNF213 (residues 1–398), which is not resolved in any of the current cryo-EM maps.

The cryo-EM model predicts that deletion of the RNF213 RING domain abolishes the interaction between RNF213 and IpaH1.4/2.5. Indeed, in a flow-cytometry-based cellular assay (Fig. 4g and Extended Data Fig. 3d), we observed that RNF213-ΔRING is no longer degraded in cells infected with *S. flexneri*, which confirms that the RING domain is the relevant binding site in cells. Because the catalytic activity of RNF213 toward LPS is mediated by the RZ domain[11], binding of IpaH1.4/2.5 to the distal RING domain explains why binding alone does not inhibit the activity of RNF213 (Fig. 2b).

Interestingly, IpaH1.4 targets at least two E3 ligases (RNF213 and the LUBAC subunit HOIP), which are involved sequentially in the antibacterial ubiquitylation cascade[11,12,41]. Because IpaH1.4 recognizes both

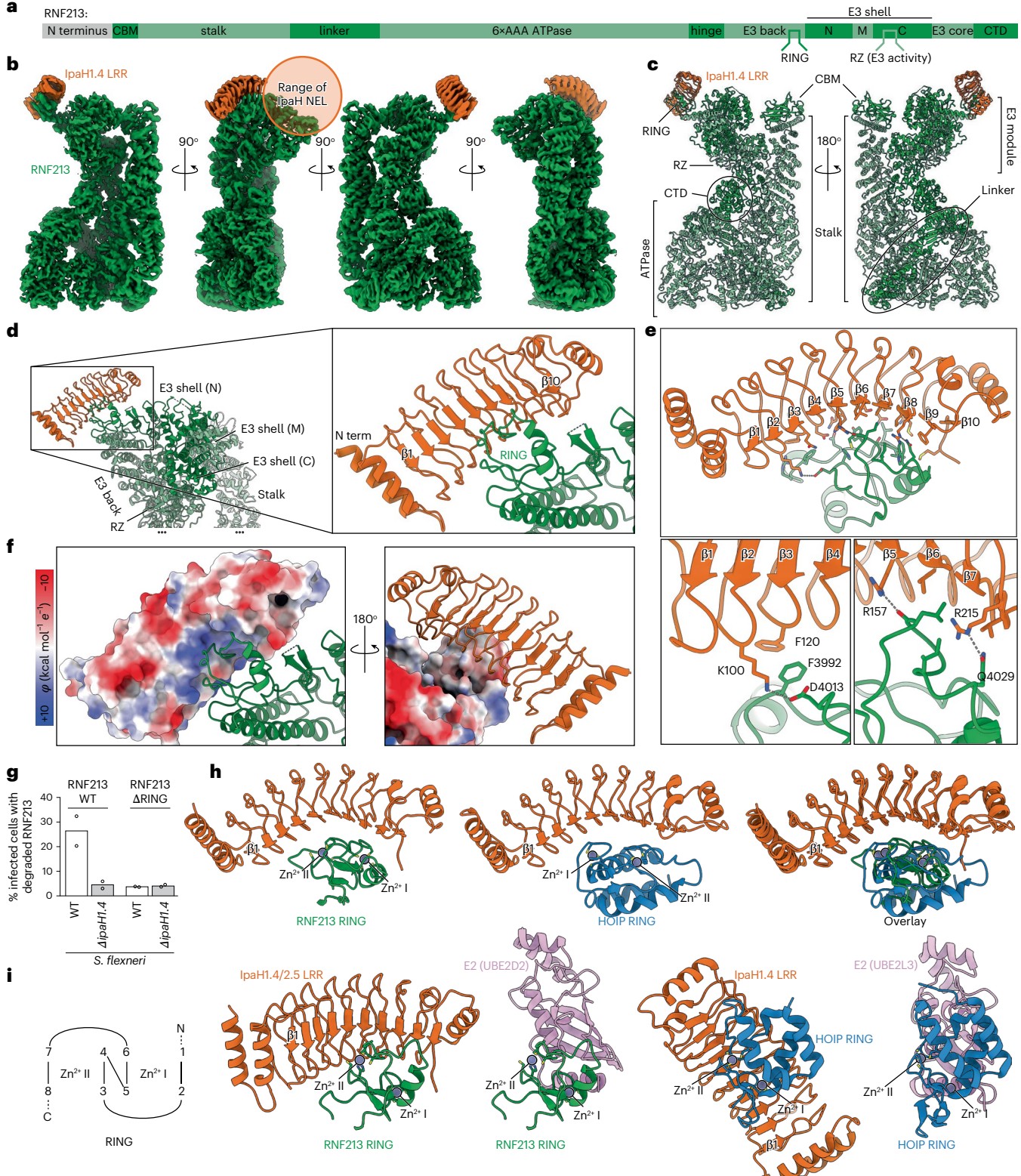

**Fig. 4 | Structure of the IpaH1.4–RNF213 complex. a**, Domain annotation of human RNF213. **b,c**, Cryo-EM map of the IpaH1.4–RNF213 complex at 2.9–3.3-Å resolution (**b**) and corresponding atomic model (**c**), all colored by subunit: IpaH1.4, orange; RNF213, green. **d**, Magnified view of the interaction interface between the IpaH1.4 LRR and the RNF213 RING domain. **e**, Key interactions between IpaH1.4 and RNF213. **f**, The interacting surface of IpaH1.4 with RNF213, shown from two opposing views, colored by electrostatic potential ($\varphi$). **g**, Degradation of RNF213 and RNF213-ΔRING by *S. flexneri* WT and *ΔipaH1.4*, quantified by flow cytometry (Extended Data Fig. 3d). Bars denote the mean from

$n$ = 2 experiments performed in duplicate. **h**, Structure comparison of IpaH1.4 LRR–RNF213 RING (this work) and IpaH1.4 LRR–HOIP RING1 (PDB 7V8G), aligned by the IpaH1.4 LRR. **i**, Structure comparison of IpaH1.4 LRR–RNF213 RING (this work), UBE2D2–RNF213 RING (predicted), IpaH1.4 LRR–HOIP RING1 and UBE2D2–HOIP RING1 (PDB 7V8G and 7V8F)[42], all aligned by the Zn fingers in the RING domains and colored by subunit: IpaH1.4, orange; RNF213, green; UBE2D2, pink; HOIP, blue. The Zn fingers are labeled I and II as per the shown domain architecture diagram (1–8 denote the Zn-coordinating residues, cysteine or histidine).

**Table 1 | Cryo-EM data collection and processing, model refinement and validation statistics**

| | RNF213–IpaH1.4 | RNF213–IpaH2.5 |
|---|---|---|
| EM Data Bank | EMD-50913 | EMD-50914 |
| PDB | 9G08 | 9G09 |
| **Data collection** | | |
| Microscope | Titan Krios | Titan Krios |
| Specimen temperature (K) | ~80 | ~80 |
| Voltage (kV) | 300 | 300 |
| Camera | Falcon 4i | Gatan K3 |
| Energy filter | Selectris X | Gatan BioQuantum |
| Energy filter slit width (eV) | 10 | 20 |
| Electron fluence ($e^-$ per $Å^2$) | 32 | 40 |
| Electron flux ($e^-$ per pixel per s) | 6.0 | 15.6 |
| Exposure time (s) | 4.56 | 1.75 |
| Magnification | ×130,000 | ×105,000 |
| Pixel size (Å) | 0.921 | 0.825 |
| Defocus range (μm) | 0.7–3.0 | 0.5–3.5 |
| Average defocus (μm) | 1.9 | 2.1 |
| Total number of movies | 16,931 | 23,344 |
| **Data processing** | | |
| Initial no. of particle images | 859,205 | 646,382 |
| Final no. of particle images | 334,921 | 213,053 |
| Particle box size (pixels) | 384×384 at 1.228 Å per pixel | 400×400 at 1.2375 Å per pixel |
| Symmetry imposed | $C_1$ | $C_1$ |
| Map resolution at gold-standard FSC=0.143 cutoff (Å) | 3.3 | 3.4 |
| Local resolution range (Å) | 2.8–4.5 | 2.9–4.5 |
| **Refinement** | | |
| Initial model used (PDB code) | 8S24, 7YA7 | 8S24, 7YA8 |
| Refinement package | Coot, ISOLDE, PHENIX | Coot, ISOLDE, PHENIX |
| Model resolution at map-model FSC=0.5 cutoff (Å) | 3.4 | 3.1 |
| Map-sharpening $B$ factor ($Å^2$) | 66–104 | 72–96 |
| Model composition | | |
| Nonhydrogen atoms | 38,432 | 38,460 |
| Protein residues | 4,766 | 4,766 |
| Ligands | 3× Zn, Mg, ATP | 3× Zn, Mg, ATP |
| Molecular weight (kDa, incl H) | 547 | 547 |
| $B$ factors ($Å^2$) | | |
| Protein | 92 | 44 |
| Ligand | 82 | 60 |
| Root-mean-square deviations | | |
| Bond lengths (Å) | 0.005 | 0.005 |
| Bond angles (°) | 0.6 | 0.7 |
| **Validation** | | |
| MolProbity score | 1.9 | 1.9 |
| Clashscore | 5.6 | 5.5 |
| Poor rotamers (%) | 2.2 | 2.1 |
| Cβ deviations (%) | 0.0 | 0.0 |
| CaBLAM outliers (%) | 2.0 | 1.7 |
| Ramachandran plot | | |
| Favored (%) | 94.5 | 95.0 |
| Allowed (%) | 5.3 | 4.9 |
| Outliers (%) | 0.2 | 0.1 |

FSC, Fourier shell correlation.

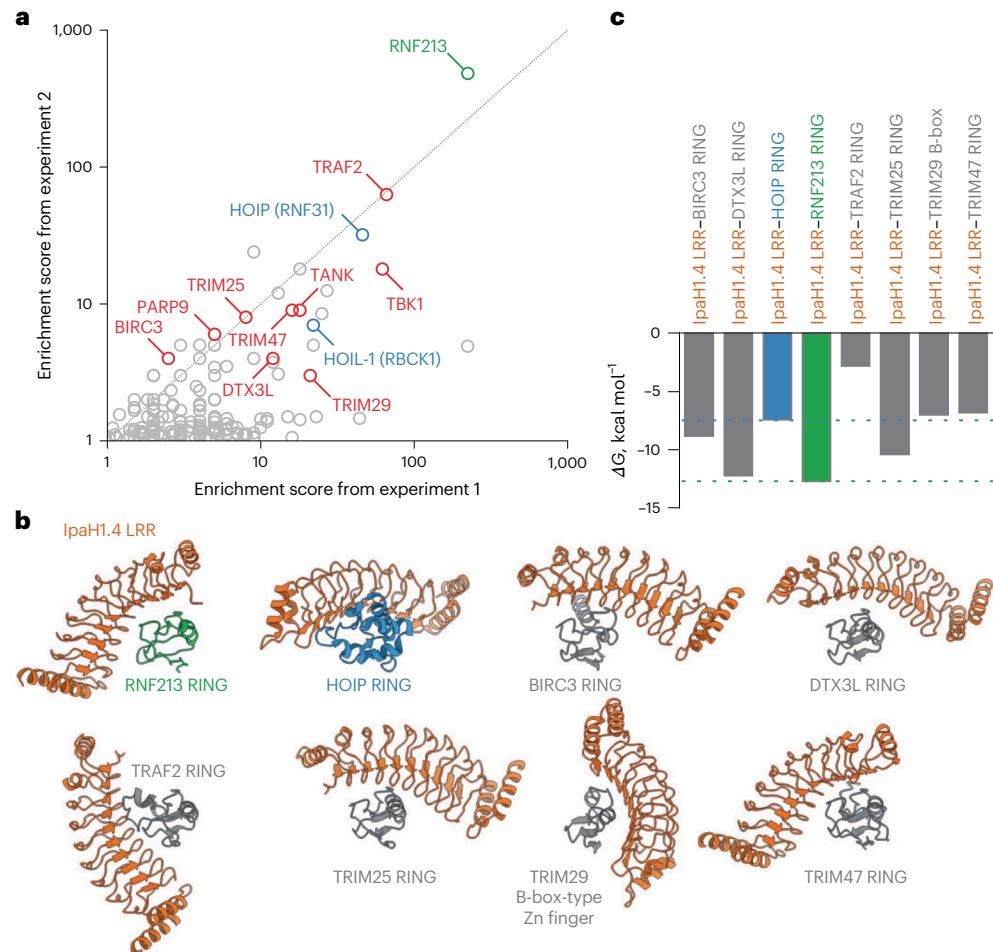

**Fig. 5 | MS-based identification of potential RING domain-containing targets of IpaH1.4/2.5 in human cells. a**, Truncated scatter plot of proteins enriched in pulldowns with GST–IpaH1.4/2.5 relative to negative controls (GST and GST–IpaH9.8). The enrichment score is defined as the ratio (summed number of peptides detected in the IpaH1.4/2.5 samples + 1)/(summed number of peptides detected in the control samples + 1). Enrichment scores from two separate experiments are plotted; only proteins with an enrichment score > 1.0 in both experiments are shown. A complete list is provided in Supplementary Table 1. RNF213 is in green, LUBAC subunits HOIL-1L and HOIP are in blue, other RING or RING-like domain-containing proteins and their known binding partners are in red (for example, PARP9 and DTX3L (ref. 57) and TRAF2, TBK1 and TANK

(ref. 58)). **b**, Models of the IpaH1.4 LRR interaction with the RING domains of RNF213 and HOIP (experimental) and of BIRC3, DTX3L, TRAF2, TRIM25, TRIM29 and TRIM47 (predicted). The predicted models were obtained from full-length protein sequences using AlphaFold 3. Only the interacting domains are shown for clarity. All structures are displayed with their RING and B-box domains in identical orientation. **c**, Binding energies ($\Delta G$) for experimentally determined (colored) and predicted (gray) interfaces, calculated using PDB PISA[59]. Dashed lines, $\Delta G$ range for the two experimentally determined interfaces (IpaH1.4 LRR with RNF213 RING and with HOIP RING). Three of the six predicted interfaces have binding energies within this range.

HOIP and RNF213 through their RING domains (Fig. 4h)[42], the question arises whether IpaH1.4 may target additional RING domain-containing proteins and how specificity is conferred. We compared the structures of IpaH1.4 bound to the RING domain of the LUBAC subunit HOIP known as RING1 (Protein Data Bank (PDB) 7V8G)[42] or the RING domain of RNF213 reported here by superimposing IpaH1.4 and the respective RING domains (Fig. 4h,i). This comparison reveals that binding of IpaH1.4 to RNF213 and HOIP is mutually exclusive and that, in either complex, IpaH1.4 engages and occludes the E2-binding face of the RING domains. However, the respective RING domains bind the IpaH LRR in different orientations, with $Zn^{2+}$ sites I and II of the RNF213 RING domain aligning with sites II and I of the LUBAC RING1 domain, respectively, when the structures of the two complexes are aligned by the IpaH1.4 LRR (Fig. 4h). These two binding modes are mediated through interactions with distinct LRR residues, such that only IpaH1.4 K100 and R157 appear engaged with both RING substrates (Extended Data Fig. 8b)[42], although the residues with which they interact are not topologically equivalent in the two RING domains. We conclude

that the distinct binding mode of the HOIP and RNF213 RING domains reveals notable substrate specificity in IpaH1.4/2.5, a finding that argues against the possibility of IpaH1.4/2.5 globally targeting RING domain-containing proteins.

However, we nevertheless expect that IpaH1.4/2.5 might be able to engage select additional RING E3 ligases through their E2-binding interfaces. We, therefore, performed an unbiased pulldown mass spectrometry (MS) screen for IpaH1.4/2.5 interactors from HeLa cell lysates (Fig. 5a and Supplementary Table 1). We also tested lysates from interferon-γ-treated cells because IpaH-family effectors are known to associate with certain interferon-induced host proteins, such as GBPs[34,35]. In addition to RNF213 and HOIP, we found five other RING domain-containing proteins specifically enriched in the IpaH1.4 and/or IpaH2.5 pulldowns (namely, TRIM25, TRIM47, DTX3L, TRAF2 and BIRC3), as well as the B-box-type Zn finger-containing E3 ligase TRIM29. Remarkably, all of these E3 ligases have been implicated in inflammatory signaling and immunity. Whether and how any of these proteins are targeted by the *Shigella* effectors during infection remains

to be determined. Computational structure predictions position the IpaH1.4/2.5 LRR domain at the E2-binding interface of each potential interactor, although the relative orientation of the domains varies between individual complexes (Fig. 5b,c). We conclude that the *Shigella* effectors IpaH1.4/2.5 have evolved to target a limited number of select RING E3 ligases involved in immune signaling, in addition to RNF213 and LUBAC, as part of the *Shigella* strategy to manipulate host immunity.

## Discussion

Here, we report that *S. flexneri* uses the secreted effector IpaH1.4 to bind, ubiquitylate and target the host E3 ligase RNF213 for degradation by the proteasome, thereby evading LPS ubiquitylation in the cytosol of infected host cells. In the ensuing arms race between pathogen and host, the host seeks to ubiquitylate and target the pathogen for degradation, while the pathogen aims to ubiquitylate and degrade the host ubiquitylation machinery. IpaH1.4 represents the first known example of a protein from a cytosol-dwelling pathogen to directly antagonize RNF213-mediated LPS ubiquitylation. In contrast to such direct antagonism by IpaH1.4, *Burkholderia thailandensis* and certain *Chromobacterium* species deploy the deubiquitylase TssM to reverse rather than prevent ubiquitylation of LPS[48]. Yet another bacterial anti-RNF213 strategy is exemplified by the *Chlamydia trachomatis* effector GarD, a putative transmembrane protein, which prevents RNF213 recruitment to the pathogen-containing vacuole in *cis*, but its mechanism remains to be understood[17]. The fact that RNF213 is targeted directly and indirectly by bacterial effector proteins from distinct species and through a variety of strategies highlights the evolutionary importance of RNF213 in restricting intracellular pathogens. Whether further strategies against RNF213-mediated ubiquitylation have evolved in other cytosol-dwelling pathogens should be investigated as it may aid in elucidating the molecular mechanisms of RNF213 and in the development of novel therapeutic strategies.

Interestingly, IpaH1.4 antagonizes antibacterial ubiquitylation on multiple levels. In addition to preventing LPS ubiquitylation through RNF213 degradation as reported here, IpaH1.4 blocks downstream M1-linked ubiquitin chain formation by LUBAC in an analogous manner (that is, through binding the E2-interacting face of the HOIP RING1 domain and subsequent ubiquitylation of the host ligase)[12,41]. We previously showed that, at least in the context of cytosolic *Salmonella*, LPS ubiquitylation by RNF213 is required for the recruitment of LUBAC to the pathogen and further M1-linked ubiquitin chain formation[12]. Therefore, a single bacterial effector, IpaH1.4, has evolved to target multiple host enzymes that are all part of the same antibacterial ubiquitylation pathway. In addition to RNF213 and LUBAC, further immune-related RING E3 ligases may be targeted by the *Shigella* effector IpaH1.4. The putative candidates, shown in Fig. 5a, include the adenosine diphosphate (ADP)-ribosylation complex DTX3L–PARP9 with the tantalizing ability to ubiquitylate nucleic acids and ADP-ribosylated substrates in vitro[49–51] and the ISG15 E3 ligase TRIM25 (ref. 52), among others. Their interactions with the *Shigella*-secreted effectors remain to be characterized further and may offer new insights into their roles in antibacterial innate immunity. Nevertheless, deletion of IpaH1.4, either alone or in combination with IpaH2.5, was insufficient for RNF213-mediated ubiquitylation to restrict *Shigella* replication in a cell-based infection model, a situation reminiscent of TssM in *Burkholderia thailandensis*, whose deletion also did not affect bacterial replication in response to bacterial ubiquitylation[48]. Such lack of phenotype is often caused by the presence of additional inhibitors, particularly in pathogens with a plethora of effectors, including *Legionella pneumophila* and *Shigella*, where host pathways are simultaneously targeted by multiple effectors (for example, IpaH9.8 and OspC3, which inhibit GBPs and caspase 4, respectively)[35,53]. The specific mechanisms that ubiquitylated *S. flexneri ΔipaH1.4* deploys to evade autophagy remain to be elucidated and may reveal hitherto unknown bacterial effector–host protein pairs.

Lastly, we would like to point out that both RNF213 and IpaH1.4 are unconventional E3 ubiquitin ligases with poorly understood catalytic mechanisms. The E3 ligase activity of RNF213 and IpaH-family effectors are found in their RZ and NEL domains, respectively[11,27]. The structural basis for *trans*-thiolation catalysis by either of these domains has not yet been visualized. In the structure of IpaH1.4/2.5 in complex with RNF213 presented here, we did not resolve the NEL domain of the bacterial ligase, despite extensive classification and local refinements, which is likely because of the flexible linker between the LRR and the NEL and is consistent with the propensity of the NEL domain to engage with a multitude of nearby ubiquitin acceptors. RNF213 is unusual in harboring two putative E3 ligase domains: an RZ finger, whose E3 ubiquitin ligase function is well established, and a RING domain, whose function remains elusive. The RNF213 RING domain is atypical in that it lacks the 'linchpin' arginine or lysine residue normally required to catalyze ubiquitin transfer from a charged E2 enzyme[54] and it appears to contribute neither to the autoubiquitylating activity of RNF213 in vitro (Extended Data Fig. 2a) nor to the ubiquitylation of LPS by RNF213 (ref. 11). However, missense mutations impacting the RNF213 RING finger in patients with Moyamoya disease strongly suggest functional importance to the domain[55,56]. We anticipate that the IpaH1.4 LRR domain may become a useful biochemical tool to further the study of the RNF213 RING domain. The fact that the IpaH1.4-binding interface of the RING domain, despite the evolutionary pressure exerted by *Shigella* IpaH1.4, has not evolved away suggests that the RNF213 RING has some important function and that RNF213-binding partners in the host cell remain to be discovered.

## Online content

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

## Methods

### Plasmids, antibodies and reagents

The open reading frames encoding IpaH-family proteins were cloned into M6P plasmids as N-terminal GFP fusions; these were used to produce recombinant murine leukemia viruses for stable protein expression in mammalian cells[60]. For recombinant protein expression in *Escherichia coli*, the same genes were cloned into pETM-30 plasmids. Catalytically inactive variants of the IpaH proteins were produced by substituting the catalytic cysteine to alanine, as previously described[35]. Briefly, the following mutant IpaH proteins were encoded as catalytically inactive variants: *ipaH1.4*[C368A], *ipaH2.5*[C368A], *ipaH7.8*[C357A] and *ipaH9.8*[C337A].

The following primary antibodies were used in this work: anti-conjugated ubiquitin (FK2) (Enzo Life Science, BML-PW8810), anti-GroEL (Enzo Life Science, ADI-SPS-875-F), anti-actin (Abcam, ab8227), anti-RNF213 (Merck, HPA003347 and HPA026790), anti-Flag–M2–HRP (horseradish peroxidase; Merck, A8592), anti-GFP (JL8, Clontech, 632381), anti-K48-linked ubiquitin (Abcam, ab140601), anti-*Salmonella* Typhimurium LPS (Bio-Rad, 8210-0407) and *Salmonella* O antiserum group B (BD Difco, 229481). The secondary antibodies used were from Thermo Fisher Scientific (Alexa-conjugated anti-mouse, anti-goat, anti-human and anti-rabbit antisera) and Dako (HRP-conjugated reagents).

### Cell culture

HeLa, 293ET and mouse embryonic fibroblasts (MEFs) were grown in Iscove's modified Dulbecco's medium (IMDM) supplemented with 10% fetal calf serum (FCS) at 37 °C in 5% CO₂. Cell lines were not authenticated. All cell lines were tested and found to be negative for *Mycoplasma*. Stable cell lines were generated by retroviral transduction, except for the Flag–GFP–RNF213 lines, which were generated using an inducible PiggyBac transposon system, as previously described[11]. The *RNF213*-KO cell lines used were also previously described[11]. HeLa cells, recombinantly overexpressing IpaH proteins (WT and catalytically dead variants) were generated by viral transduction, followed by antibiotic selection.

### Bacteria

*E. coli* strains MC1601 and NEB 10-beta were used for plasmid production and the BL21(DE3) strain was used for protein purification. All *E. coli* strains were grown on tryptic yeast extract (TYE) agar plates or in Luria broth (LB) or Terrific broth (TB). *S. enterica* serovar Typhimurium strain 12023 and the isogenic *ΔrfaL* were provided by D. Holden. *Salmonella* Typhimurium was grown in LB or on LB agar plates. *S. flexneri* strain M90T (WT and *ΔrfaL*) was provided by C. Tang (University of Oxford), whereas strains M90T *ΔmxiE* (ref. 61) and *ΔipaH9.8* (ref. 61) were provided by J. Rohde (Dalhousie University); strains *ΔipaH1.4*, *ΔipaH2.5* and *ΔipaH1.4/2.5* were described previously[41]. *S. flexneri* was grown in tryptic soy broth (TSB) or on tryptic soy agar containing 0.003% Congo red.

### Bacterial infections

The indicated strains of *Salmonella* Typhimurium were grown overnight in LB and subcultured (1:33) in fresh LB for 3.5 h before infection. The indicated strains of *S. flexneri* were grown overnight in TSB and subcultured (1:100) in fresh TSB for 2.5 h before infection. Host cells for infection were seeded in 24-well or 6-well plates or 10-cm dishes and cultured in antibiotic-free IMDM supplemented with 10% FCS for 1 h before infection. To infect cells with *Salmonella* Typhimurium, 500 µl of bacterial subculture was added per 10-cm dish or the linearly scaled volume for cells in multiwell plates. To infect cells with *S. flexneri*, the bacterial subculture was first washed in PBS and then resuspended in antibiotic-free cell medium without dilution. Next, 2 ml of the bacterial suspension was added per 10-cm dish and the dishes were centrifuged at 845*g* for 10 min at room temperature.

The infected cells, in both cases, were incubated at 37 °C for 30 min. After that, the cells were washed two times with warm PBS and cultured in 100 µg ml⁻¹ gentamycin (after *Salmonella* Typhimurium infection) or 50 µg ml⁻¹ gentamycin (after *S. flexneri* infection). Infected cells were harvested 4 h after infection. The cells were washed once with PBS and lysed in 1 ml of lysis buffer (PBS, 0.1% Triton X-100, 2 mM iodoacetamide and protease inhibitors) at 4 °C. The cell lysates were centrifuged at 400*g* for 5 min at 4 °C and the supernatant containing the postnuclear cell lysate and the bacteria was collected and centrifuged at 16,100*g* for 10 min at 4 °C. The supernatant containing the clarified host cell lysate was saved and the bacterial pellet was washed once with PBS, followed by bacterial lysis in 35 µl of BugBuster (Merck) including 2 mM iodoacetamide for 5 min at room temperature with occasional vortexing. The bacterial lysate was centrifuged at 16,100*g* for 10 min at room temperature and the supernatant (clarified bacterial lysate) was split into two fractions. One part was directly mixed with Laemmli buffer (Bio-Rad) containing 100 mM DTT and boiled for 5 min. The remainder of the clarified bacterial lysate was used to further purify ubiquitylated LPS. It was heated at 100 °C for 15 min and centrifuged at 16,100*g* for 10 min at room temperature; this heat-cleared supernatant was mixed with Laemmli buffer containing 100 mM DTT and boiled for 5 min.

### Immunoblot analysis

Postnuclear supernatants from HeLa or MEF cells, bacterial lysates from these infected cells and bacterial LPS from these infected cells were obtained as described above. Heat-cleared bacterial lysates were used to visualize ubiquitylated LPS with FK2 antibody, non-heat-cleared bacterial lysates were used to probe for GroEL as a loading control and postnuclear HeLa or MEF lysates were used to probe for host proteins. All samples were mixed with Laemmli buffer containing 100 mM DTT and boiled for 5 min. Samples were run on NuPAGE 4–12% Bis–Tris gels (Thermo Fisher Scientific) in MOPS SDS running buffer at 180 V for 70 min. Overnight wet transfer at 20 V was used to transfer proteins onto methanol-activated PVDF membranes (Millipore). Membranes were blocked for 1 h in PBS-T (137 mM NaCl, 2.7 mM KCl, 10 mM Na₂HPO₄, 1.8 mM KH₂PO₄ and 0.1% Tween 20) containing 5% BSA, followed by overnight incubation at 4 °C or 1–4 h incubation at room temperature with primary antibodies in PBS-T containing 5% BSA. Membranes were washed in PBS-T, incubated with secondary HRP-conjugated antibodies in PBS-T containing 5% milk for 1 h at room temperature, followed by three PBS-T washes. Visualization following immunoblotting was performed using enhanced chemiluminescence detection reagents (Amersham Bioscience) and a ChemiDoc MP imaging system (Bio-Rad). Alternatively, instead of HRP-conjugated antibodies, fluorophore-conjugated antibodies were used in some instances, in which case the blots were visualized directly after the final wash.

### Microscopy of fixed cells

Cells were grown on glass coverslips and, after infection and fixation in 4% paraformaldehyde, were washed twice in PBS, permeabilized for 5 min in PBS with 0.1% Triton X-100 and blocked in PBS with 2% BSA for 1 h. Cells were incubated overnight with primary antibodies followed by Alexa-conjugated secondary antibodies in blocking solution for at least 1 h at room temperature. Coverslips were then mounted in Prolong gold mounting medium. Images of infected cells were acquired on a Zeiss 780 inverted microscope using a ×63/1.4 numerical aperture (NA) oil immersion lens. Images were analyzed using Fiji[62]. Marker-positive bacteria were scored by eye from coverslips on a Zeiss Axio Imager microscope using a ×100/1.4 NA oil immersion lens, with at least 200 bacteria per experiment. In the case of *Salmonella* Typhimurium, only large (cytosolic) bacteria were considered for quantification.

### Enumeration of intracellular bacteria

A slight variation on the above protocol for bacterial infection was used to infect cells for enumeration of intracellular bacteria. *S. flexneri*

was grown overnight in TSB and subcultured (1:100) in fresh TSB for 2.5 h before infection. Such cultures were consecutively washed in PBS and resuspended in antibiotic-free IMDM supplemented with 10% FCS immediately before using 10 µl to infect HeLa or 20 µL for MEF cells in 24-well plates. The plates with infected cells were centrifuged for 10 min at 670$g$ followed by incubation at 37 °C for 15 min. Following two washes with warm PBS, cells were cultured in 100 µg ml⁻¹ gentamycin for 1 h and 20 µg ml⁻¹ gentamycin thereafter. To enumerate intracellular bacteria, cells from triplicate wells were lysed in 1 ml of PBS containing 0.1% Triton X-100. Serial dilutions were performed in PBS and plated in duplicate on TYE agar.

### Flow cytometry

A slight variation in the above protocol for bacterial infection was used to infect cells for analysis by flow cytometry. MEF cells harboring expression cassettes for doxycycline-inducible GFP–RNF213 (ref. 11) or GFP alone were cultured in 24-well plates for 48 h before infection with the indicated strains of *S. flexneri*. GFP–RNF213 or GFP expression was induced by the addition of 1 µM doxycycline to antibiotic-free IMDM supplemented with 10% FCS 18 h before infection. The cells were washed twice in warm PBS and infected with 100 µl of *S. flexneri* in IMDM supplemented with 10% FCS for 30 min in triplicate wells, as described above. To test the effects of autophagy and proteasome inhibitors, the cells were treated with 100 nM bafilomycin A1 or carfilzomib, respectively, using DMSO as a negative control. The ligands were added 15 min before infection and maintained throughout the experiment. Cells were cultured in 100 µg ml⁻¹ gentamycin for 1 h after infection and 20 µg ml⁻¹ gentamycin thereafter. When testing the GFP–RNF213-ΔRING variant, the cells were also treated with the pan-caspase inhibitor Q-VD (20 µM) at the time of doxycycline induction to prevent RNF213-ΔRING-induced cell death. Q-VD was removed before infection. For the complementation of *S. flexneri* Δ*ipaH1.4*, full-length *ipaH1.4* was cloned into the plasmid p4928 (Addgene, 168177; gift from M. Hensel)[63]. Expression of *ipaH1.4* was induced by addition of 100 ng ml⁻¹ anhydrotetracycline to the *S. flexneri* subculture 1 h before infection and maintained throughout the experiment. For the complementation of *S. flexneri* Δ*ipaH1.4* with mutant alleles, the *ipaH1.4* sequences, followed by a C-terminal HA tag, were cloned into the plasmid pFPV, which also encodes a fluorescent reporter (mRuby) from a bicistronic transcript. Infected cells were washed with PBS and trypsinized at either 1 h or 5 h after infection, centrifuged at 400$g$ for 2 min and washed twice with PBS. Cells were fixed in 4% paraformaldehyde for 15 min, washed twice in PBS and resuspended in PBS supplemented with 1% BSA.

For detection of endogenous RNF213 by flow cytometry, HeLa WT or *RNF213*-KO cells were infected with the indicated strains of *S. flexneri* for 30 min and trypsinized at 4 h after infection. Cells were pelleted at 400$g$, washed once in PBS and subsequently fixed and permeabilized using a cell fixation and permeabilization kit (Nordic MUbio). Anti-RNF213 antibody (HPA026790) was included during permeabilization for 45 min. Cells were washed twice in PBS supplemented with 5% FCS and incubated with Alexa-fluor 488-conjugated anti-rabbit secondary antibody at 1:500 dilution in permeabilization buffer for 30 min. Cells were washed twice and resuspended in PBS supplemented with 5% FCS. All samples were run on a LSRFortessa flow cytometer, collecting a total of 30,000 events per sample. The quantitative difference in staining of RNF213 WT and KO cells was interpreted as the specific contribution of RNF213 to the assay.

For bacterial spreading experiments cells were infected with a relatively low multiplicity of infection of Ruby-expressing *S. flexneri* to achieve 2–5% infected cells at 1 h after infection as assessed by flow cytometry. This would presumably permit each individual infected cell to spread *S. flexneri* to neighboring uninfected cells. Infection was allowed to proceed for 5 h at which point the percentage of infected cells was quantified and the fold spread was calculated as

(percentage infected at 5 h)/(percentage infected at 1 h). All data were analyzed in FlowJo (BD Biosciences).

### FACS

Live, rather than fixed, infected cells were sorted using a BD FACSAria Fusion fluorescence-activated cell sorter. HeLa cells were infected with the indicated strains of *S. flexneri* as described above. The cells were harvested by scraping into PBS at 4 h after infection and washed twice with PBS. The cells were sorted into three populations on the basis of the *Shigella*-encoded mRuby fluorescence (uninfected, low fluorescence and high fluorescence), where the fluorescence intensity of each cell served as a proxy for bacterial load in the cell. For each specimen, 10,000 cells were sorted into each of the three populations. The collected cells were pelleted by centrifugation and resuspended in 10 µl of lysis buffer (8 M urea, 50 mM Tris-HCl pH 8, 0.5% NP40 and 0.1% SDS), supplemented with 1:100 universal nuclease (Pierce). The cell lysates were mixed with Laemmli buffer, boiled and subjected to immunoblot analysis as described above.

### Detection of IpaH1.4/2.5 transcripts in infected cells

MEF *RNF213*-KO cells were infected with *S. flexneri* for 30 min, as described above, washed twice in warm PBS and cultured for a further 30 min. Cells were lysed in buffer RLT (Qiagen) with the addition of 143 mM β-mercaptoethanol and RNA was extracted using an RNeasy extraction kit (Qiagen). Complementary DNA was synthesized using Superscript III reverse transcriptase (Thermo Fisher Scientific) and random hexamer primers from 1.5 µg of RNA per sample. Real-time Taqman qPCR was performed using primers specific for *ipaH1.4* (ATCGGTTCTG-GCATTATGCAT and TCATTACAAGATGGAGTCAGTTCCA) or *ipaH2.5* (CGGTTCTGGTATTATGCATCAAAT and CTCAAACTCACAGGCAAA-GAAAATAG) and FAM–MGB-labeled probes (AACAATGTATACTCGT-TAACTC for *ipaH1.4* and AACAACATACACTCGTTAAC for *ipaH2.5*). Primers and probe for the murine *Actb* gene (Mm01205647_g1, Thermo Fisher Scientific) were used to normalize data across samples. Plasmids harboring *ipaH1.4* or *ipaH2.5* were used to quantify the number of molecules of each per infected well amounting to approximately 10⁵ cells. Cells infected with either *S. flexneri* Δ*ipaH1.4* or Δ*ipaH2.5* were used to demonstrate the specificity of the primers and probes for either gene.

### Protein production in *E. coli*

IpaH1.4/2.5 were recombinantly overexpressed in *E. coli* BL21(DE3) as 6×His–GST N-terminal fusions, both in the pETM-30 vector. The plasmids were transformed into BL21(DE3) cells. Overnight cultures, started from a single colony, were used to inoculate 2 L of TB (supplemented with 50 µg ml⁻¹ kanamycin) at a ratio of 1:200. The cultures were grown at 37 °C and 200 rpm until reaching an optical density at 600 nm of 2 and then induced with 1 mM IPTG for 20 h at 200 rpm, 18 °C. Following collection by centrifugation, the cell pellets were resuspended in 200 ml of lysis buffer (20 mM Tris-HCl pH 8 and 300 mM NaCl), supplemented with 0.5 mM TCEP, protease inhibitor (cOmplete EDTA-free tablets, Roche), lysozyme (1 mg ml⁻¹, Sigma), and 5 µl of universal nuclease (Pierce). Cells were disrupted using a high-pressure homogenizer at 38,000 psi and 4 °C. Cell debris was removed by centrifugation at 30,000$g$ for 30 min at 4 °C and the volume of the supernatants was adjusted to 350 ml each by adding lysis buffer. Proteins were purified by immobilized metal-affinity chromatography. The cell lysates were mixed with 2.5 ml of Ni-NTA Agarose resin (Qiagen). Binding to the resin was allowed to proceed for 2 h while shaking at 110 rpm at 10 °C. The beads were then collected by gravity flow and washed three times with 50 ml of of wash buffer (25 mM Tris-HCl pH 8, 400 mM NaCl and 0.5 mM TCEP), containing increasing imidazole concentrations of 25 mM, 37.5 mM, and 50 mM. The proteins were eluted in 20 ml of of elution buffer (25 mM Tris-HCl pH 8, 400 mM NaCl, 0.5 mM TCEP and 500 mM imidazole) and concentrated to a final volume of ~2 ml. The 6×His–GST tag was removed by incubation

with His-tagged tobacco etch virus protease at 4 °C for 13 h in cleavage buffer (50 mM Tris-HCl pH 8, 150 mM NaCl, 0.5 mM EDTA and 0.5 mM TCEP). Untagged IpaH1.4/2.5 were further purified either by reverse immobilized metal-affinity chromatography (for use in cryo-EM) or by anion-exchange chromatography using a Resource Q column (Cytiva) in buffer containing 20 mM Tris-HCl pH 8.5, 10–500 mM NaCl and 4 mM DTT and size-exclusion chromatography using a Superdex 200 16/60 column (Cytiva) in buffer containing 20 mM Tris-HCl pH 8.5, 200 mM NaCl and 4 mM DTT (for all other experiments).

### Protein production in *Spodoptera frugiperda*

WT human RNF213 was expressed in insect cells and purified as previously described[11,14]. Briefly, pOP806_pACEBac1 2×Strep–RNF213 plasmid was transformed into DH10EmBacY cells (DH10Bac with yellow fluorescent protein reporter). Blue–white screening was used to isolate colonies containing recombinant baculoviral shuttle vectors (bacmids) and bacmid DNA was extracted combining cell lysis and neutralization using buffer P1, P2 and N3 (Qiagen), followed by isopropanol precipitation. A six-well plate of *S. frugiperda* cells (Sf9 cells; Oxford Expression Technologies) grown at 27 °C in Insect-Xpress media (Lonza) without shaking was transfected with bacmid plasmid using PEI transfection reagent. After 6 days, virus P1 was collected and used 1:25 to transduce 50 ml ($1.8 \times 10^6$ cells per ml) of Sf9 cells. After 7 days of incubation at 27 °C with 140 rpm shaking, virus P2 was collected. To express protein, 1 L ($2.6 \times 10^6$ cells per ml) of Sf9 cells were transduced with a 1:50 dilution of P2 virus and incubated at 27 °C with 140 rpm shaking for 72 h. Cells were pelleted by centrifugation, snap-frozen in liquid nitrogen and stored at −80 °C. To lyse cells, the pellets were thawed and resuspended to a total volume of 180 ml in lysis buffer (30 mM HEPES, 100 mM NaCl, 10 mM $MgCl_2$ and 0.5 mM TCEP, pH 7.6) containing 20 µl of universal nuclease (Pierce), 20 µl of benzonase nuclease (Sigma Aldrich) and two EDTA-free protease inhibitor tablets (Roche). The cell suspension was stirred for 1 h and then sonicated for 50 s in 5-s pulses with a 25-s waiting time at 70% amplitude using a 130-W microtip sonicator (Vibra Cell). The lysate was centrifuged at 20,000$g$ for 60 min at 4 °C. The clarified lysate was filtered through a 0.2-µm filter (Millipore) and applied to two 5-ml StrepTrap HP columns (GE Healthcare) connected in series. After washing the columns with 100 ml of lysis buffer, RNF213 was eluted with lysis buffer supplemented with 2.5 mM desthiobiotin pH 8. The eluted protein was kept at 4 °C and used immediately for cryo-EM specimen preparation. All purification steps were carried out at 4 °C using an ÄKTA pure 25 (GE Healthcare). RNF213 variants, including RNF213-ΔRING, RNF213-H4509A and RNF213-E2488A;E2845A, were expressed and purified analogously.

### In vitro ubiquitylation assays

First, precharged E2 enzyme was prepared by incubating 10 µM UBCH5C (Boston Biochem) with 200 µM Flag–ubiquitin (Boston Biochem) and 0.2 µM UBE1 in reaction buffer (30 mM HEPES pH 7.4, 100 mM NaCl, 10 mM $MgCl_2$, 50 mM adenosine triphosphate (ATP) and 5 mM DTT) for 15 min at 37 °C. UBE1 was purified as previously described[11]. The precharged E2 reaction was diluted fivefold with the addition of 350 nM recombinantly produced RNF213 (for RNF213 autoubiquitylation experiments) or 350 nM recombinantly produced RNF213-H4509A and 2 µM recombinantly produced IpaH proteins (for IpaH-mediated ubiquitylation of RNF213). The reaction was incubated at 37 °C for 1 h and then further processed for immunoblot analysis or directly visualized in the gel by Coomassie staining.

### Pulldown assays

Recombinantly expressed catalytically dead IpaH proteins were purified from *E. coli* using a slight variation of the above procedure. Briefly, *E. coli* carrying the GST–IpaH (catalytically dead cysteine-to-alanine mutant) constructs in a pETM-30 plasmid were grown to an optical density at 600 nm of 0.7 and protein expression was induced with

0.1 mM of IPTG for 20 h at 200 rpm, 18 °C. Following collection by centrifugation, the cell pellets were resuspended in 10 ml of lysis buffer per 1 L of bacterial culture (20 mM Tris-HCl pH 8, 150 mM NaCl and 10 mM $MgCl_2$), supplemented with 5 mM TCEP, protease inhibitor (cOmplete EDTA-free tablets, Roche), 20 mM imidazole and 20 µg ml$^{-1}$ DNase (Sigma). Cells were disrupted by freeze–thaw cycles, followed by high-pressure homogenization at 38,000 psi and 4 °C. Cell debris were removed by centrifugation at 30,000$g$ for 30 min at 4 °C and the supernatants were bound to pre-equilibrated GST–Sepharose resin (Cytiva, 1 ml per 1 L of bacterial culture) in the presence of 0.1% Triton X-100 for 1 h at 4 °C. The resin was washed in 10 ml of lysis buffer five times and the GST-tagged proteins were eluted in 2 ml of elution buffer each (20 mM Tris-HCl pH 8, 25 mM reduced glutathione, 150 mM NaCl and 10 mM $MgCl_2$). The relative IpaH amounts were quantified from a Coomassie-stained SDS–PAGE and the volumes subsequently used for the pulldown assay were adjusted accordingly to arrange for equal amounts of GST–IpaH per sample. Between 8 and 24 µl of purified GST–IpaH protein was applied to 70 µl of GST–Sepharose resin. This was used to pull down RNF213 from cell lysates. The cell lysates were prepared from *RNF213*-KO MEFs, complemented with GFP–RNF213 (human). GFP–RNF213 expression was induced by the addition of 1 µM doxycycline 18 h before harvesting the cells. The cells were washed in PBS and collected in lysis buffer (20 mM Tris-HCl pH 8, 150 mM NaCl, 0.1% Triton X-100, 5 mM EDTA and protease inhibitor tablet). The cell lysate was applied to the GST–IpaH on beads, the beads were washed three times and the proteins were eluted from the resin in 100 µl of elution buffer as above and used for immunoblot analysis.

Pulldown assays to test prospective binding mutants of RNF213 or IpaH1.4 (Extended Data Fig. 8) were performed analogously, except for using bacterial lysates containing overexpressed GST–IpaH1.4 (WT or binding mutants) rather than purified protein. For these experiments, 50 µl of bacterial lysate was applied to 25 µl of GST–Sepharose resin. Cell lysates containing Flag–GFP–RNF213 (WT or binding mutants) or Flag–GFP control were prepared from 293ET cells transfected with the PiggyBac–RNF213 plasmid, including overnight incubation with doxycycline. Proteins were not eluted from the washed beads; instead, the bead suspension was boiled in Laemmli buffer and used for SDS–PAGE and immunoblot analysis.

### MS

Coprecipitation experiments to identify potential interactors of IpaH-family proteins were performed similarly to the pulldown assay as described above. GST beads (40 µl per specimen) were precharged with GST–IpaH proteins as described above. Then, $1 \times 10^8$ HeLa cells were lysed and the lysates were added to each aliquot of IpaH-bound beads. Protein pulldowns were performed at 4 °C with agitation for 2 h. Beads were then washed at least four times in cold wash buffer (20 mM Tris-HCl pH 7.4, 150 mM NaCl, 0.1% Triton X-100, 5% glycerol, 5 mM EDTA, 1 mM PMSF, 1 mM benzamidine, 2 µg ml$^{-1}$ aprotinin, 5 µg ml$^{-1}$ leupeptin and 1 mM DTT). Buffer was exchanged to 100 mM ammonium bicarbonate and proteins were reduced in the presence of 10 mM DTT and alkylated with 55 mM iodoacetamide in a total volume of 30 µl. Proteins were then digested on beads with 1 µg of trypsin (Promega) for 18 h at 37 °C. Peptides were subsequently acidified by adding 4 µl of 2% formic acid. The sample was spun down at 14,000$g$ for 5 min and 20 µl of supernatant was transferred into a vial. The peptide fractions (7 µl) were analyzed by nanoscale capillary liquid chromatography (LC)–MS/MS using an Ultimate U3000 high-performance LC instrument (Thermo Fisher Scientific Dionex) to deliver a flow of approximately 300 nl min$^{-1}$. A µ-Precolumn cartridge C18 Acclaim PepMap 100 (5 µm, 300 µm × 5 mm) (Thermo Fisher Scientific Dionex) trapped the peptides before separation on a C18 Acclaim PepMap100 (3 µm, 75 µm × 250 mm) (Thermo Fisher Scientific Dionex). Peptides were eluted with a 90-min gradient of acetonitrile (5–40%). The analytical column outlet was directly interfaced, through a modified nano-flow

electrospray ionization source, with a hybrid linear quadrupole ion trap MS instrument (Orbitrap LTQ Velos, Thermo Fisher Scientific). Data analysis was carried out using a resolution of 60,000 for the full MS spectrum followed by ten MS/MS spectra in the linear ion trap. MS spectra were collected over an $m/z$ range of 200–1,800. MS/MS scans were collected using threshold energy of 35 eV for collision-induced dissociation. LC–MS/MS data were then searched against the Uniprot database using Mascot (Matrix Science). Database search parameters were set with a precursor tolerance of 10 ppm and a fragment ion mass tolerance of 0.8 Da. One missed enzyme cleavage was allowed and variable modifications for oxidized methionine, carbamidomethyl and phosphorylated serine, threonine or tyrosine were included. Two biological replicates of this experiment were performed.

### Specimen preparation for cryo-EM
Recombinant RNF213 and IpaH1.4/2.5 were purified as described above. RNF213 was mixed with a fivefold molar excess of IpaH1.4/2.5 and incubated on ice for 30 min. The final concentration of proteins was around 2 µM RNF213 and 10 µM IpaH1.4/2.5. All-gold grids (UltrAuFoil R 1.2/1.3, 300-mesh, QuantiFoil Micro Tools) were used for all specimens. The grids were plasma cleaned in an atmosphere of 9:1 $Ar:O_2$ for 2 min at 70% power, 30-sccm gas flow in a Fischione 1070 plasma chamber. A humidified manual plunger of the Talmon type[64] situated in a 4 °C cold room was used to vitrify the specimens. Then, 3 µl of each specimen was applied to the foil side of the grid and blotted from the same side for 10 s using Whatman No. 1 filter paper. The grid was then immediately plunged into liquid ethane, held at 93 K in a temperature-controlled cryostat[65]. All grids were stored in liquid nitrogen until use.

### Cryo-EM data collection
Both datasets were collected at the Electron Bioimaging Center (eBIC) at the Diamond Light Source. The dataset for the RNF213–IpaH1.4 complex was collected on a Titan Krios (Thermo Fisher Scientific) electron cryomicroscope, equipped with a Schottky X-FEG operated at 300 keV and a Selectris X (Thermo Fisher Scientific) imaging filter with a post-filter Falcon 4i (Thermo Fisher Scientific) direct electron detector. The filter was used with a 10-eV slit and 4.56-s exposures, corresponding to a fluence of 32 e$^-$ per Å$^2$, were acquired in EER format at a nominal magnification of ×130,000, corresponding to a calibrated pixel size of 0.921 Å. The dataset for the RNF213–IpaH2.5 complex was collected on a Titan Krios electron cryomicroscope, equipped with a Schottky X-FEG operated at 300 keV and a BioQuantum (Gatan) imaging filter with a postfilter K3 (Gatan) direct electron detector. The filter was used with a 20-eV slit and 1.75-s exposures, corresponding to a fluence of 40 e$^-$ per Å$^2$, were acquired in TIFF format in super-resolution mode with twofold binning at a nominal magnification of ×105,000, corresponding to a calibrated pixel size of 0.825 Å. Both datasets were acquired using EPU (Thermo Fisher Scientific) with aberration-free image shift within a 6-µm range and correspond to 48 h of data collection each. Data collection parameters are summarized in Table 1.

### Cryo-EM data processing
Both datasets were processed using a combination of RELION-5.0 (ref. 66) and cryoSPARC-4.4 (ref. 67). The data processing strategy is summarized in Extended Data Fig. 7. Briefly, all cryo-EM movies were imported in RELION and motion-corrected. The EER frames for the RNF213–IpaH1.4 dataset were grouped into 36 fractions, each corresponding to 0.9 e$^-$ per Å$^2$ of irradiation. The contrast transfer functions were estimated using CTFFIND-4.1 (ref. 68). Initial particle picking was performed using the Laplacian-of-Gaussian picker in RELION with diameter range from 200 to 250 Å and minimal threshold at −2 s.d. The picked particles were extracted into 512-pixel boxes and binned by a factor of 4. The extracted particles were imported into cryoSPARC for two-dimensional (2D) and three-dimensional (3D) classification, where false picks and damaged particles were removed.

The selected particles yielded an initial reconstruction of RNF213. This was low-pass-filtered to 20 Å and used for template-based particle picking in RELION. The newly picked particles were extracted into 512-pixel boxes and binned by a factor of 2. The extracted particles were imported into cryoSPARC again for 2D and 3D classification. A selected subset of particles, which yielded a high-resolution reconstruction of RNF213, with additional low-resolution density corresponding to the IpaH1.4/2.5 LRR at partial occupancy were imported back into RELION and subjected to optical aberration refinement[69] and Bayesian polishing[70]. The particles were then subjected to 3D refinement with regularization by the Blush algorithm[71]. The efficiency of the orientation distributions was calculated using cryoEF[72]. Masks encompassing the E3 domain of RNF213 and the IpaH1.4/2.5 LRR were created and used for masked 3D classification without alignment to remove particles without IpaH1.4/2.5 bound, followed by local 3D refinement with regularization by the Blush algorithm. This was repeated iteratively with progressively tighter masks and the particles were resampled as necessary as resolution improved until the final mask encompassed only IpaH1.4/2.5 and the RNF213 RING domain. All other RNF213 domains (ATPase, stalk and CBM) were also locally refined independently, as previously described[14]. A composite map was produced from all refined domains and this was used for model building. Graphics were generated in ChimeraX[73].

### Atomic model building and refinement
The initial atomic models were created by docking the structures of human RNF213 (PDB 8S24)[14] and IpaH1.4/2.5 LRR (PDB 7YA7 and 7YA8)[74] into the composite cryo-EM maps (comprising the combination of the locally refined regions of the complex). The models were manually refined in ISOLDE[75] followed by real-space refinement in PHENIX[76] with secondary-structure restraints and Ramachandran restraints.

### Reporting summary
Further information on research design is available in the Nature Portfolio Reporting Summary linked to this article.

## Data availability
The cryo-EM maps and the refined atomic models were deposited to the EM Data Bank under accession codes EMD-50913, EMD-50914, EMD-50915, EMD-50916, EMD-50917, EMD-50918, EMD-50919, EMD-50920, EMD-50921, EMD-50922, EMD-50923, EMD-50924, EMD-50925, EMD-50926, EMD-50928 and EMD-50929 and the PDB under accession codes 9G08 and 9G09, respectively. Other data are available within the article and the Supplementary Information. Data and materials can be obtained from the corresponding authors upon request. Source data are provided with this paper.

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

## Acknowledgements

This work was supported by the Medical Research Council (MRC) as part of United Kingdom Research and Innovation (U105170648, to F.R.), the Wellcome Trust (222503/Z/21/Z, to F.R.), the German Research Foundation (SCHA 2458/1-1, to F.S.), a Swiss National Science Foundation postdoctoral mobility grant (P400PB_191083, to P.M.H.), the National Institutes of Health (AI083359, to N.M.A.) and the Welch Foundation (I-1704, to N.M.A.). We acknowledge Diamond for access and support of the cryo-EM facilities at eBIC (proposal BI31336), funded by the Wellcome Trust, MRC and Biotechnology and Biological Sciences Research Council. We also thank the MRC Laboratory of Molecular Biology cryo-EM, scientific computing, light microscopy, insect cell culture, MS and flow cytometry facilities for their support. We thank J. Schwab and D. Kimanius for helpful advice on cryo-EM data processing, A. Lane for helpful advice on primer design, C. Tang, J. Rohde and D. Holden for bacterial strains and M. Hensel for sharing plasmids.

## Author contributions

K.N., K.B.B., C.P., P.P., A.C.C., F.S., P.M.H. and E.G.O. performed the experiments and analyzed the results. N.M.A. provided the *Shigella* strains. K.N., K.B. and F.R. wrote the paper with input from all authors.

## Competing interests

The authors declare no competing interests.

## Additional information

**Extended data** is available for this paper at https://doi.org/10.1038/s41594-025-01530-8.

**Correspondence and requests for materials** should be addressed to Felix Randow.

```
A0A0H2USG1|IPA14_SHIFL    MIKSTNIQAIGSGIMHQINNVYSLTPLSLPMELTPSCNEFYLKTWSEWEKNGTPGEQRNI    60
A0A0H2USC0|IPA25_SHIFL    MIKSTNIQVIGSGIMHQINNIHSLTLFSLPVSLSPSCNEYYLKVWSEWERNGTPGEQRNI    60
                          ********.***********:;*** .:***:.*:*****:***.*****:*********
                                            N terminus                        Helix 1

A0A0H2USG1|IPA14_SHIFL    AFNRLKICLQNQEAELNLSELDLKTLPDLPPQITTLEIRKNLLTHLPDLPPMLKVIHAQF   120
A0A0H2USC0|IPA25_SHIFL    AFNRLKICLQNQEAELNLSELDLKTLPDLPPQITTLEIRKNLLTHLPDLPPMLKVIHAQF   120
                          ************************************************************
                               Helix 2            Leucine rich repeats

A0A0H2USG1|IPA14_SHIFL    NQLESLPALPETLEELNAGDNKIKELPFLPENLTHLRVHNNRLHILPLLPPELKLLVVSG   180
A0A0H2USC0|IPA25_SHIFL    NQLESLPALPETLEELNAGDNKIKELPFLPENLTHLRVHNNRLHILPLLPPELKLLVVSG   180
                          ************************************************************

A0A0H2USG1|IPA14_SHIFL    NRLDSIPPFPDKLEGLALANNFIEQLPELPFSMNRAVLMNNNLTTLPESVLRLAQNAFVN   240
A0A0H2USC0|IPA25_SHIFL    NRLDSIPPFPDKLEGLALANNFIEQLPELPFSMNRAVLMNNNLTTLPESVLRLAQNAFVN   240
                          ************************************************************

A0A0H2USG1|IPA14_SHIFL    VAGNPLSGHTMRTLQQITTGPDYSGPQIFFSMGNSATISAPEHSLADAVTAWFPENKQSD   300
A0A0H2USC0|IPA25_SHIFL    VAGNPLSGHTMRTLQQITTGPDYSGPQIFFSMGNSATISAPEHSLADAVTAWFPENKQSD   300
                          ************************************************************
                                                                     Linker

A0A0H2USG1|IPA14_SHIFL    VSQIWHAFEHEEHANTFSAFLDRLSDTVSARNTSGFREQVAAWLEKLSASAELRQQSFAV   360
A0A0H2USC0|IPA25_SHIFL    VSQIWHAFEHEEHANTFSAFLDRLSDTVSARNTSGFREQVAAWLEKLSASAELRQQSFAV   360
                          ************************************************************
                                              Novel E3 ligase (NEL)

A0A0H2USG1|IPA14_SHIFL    AADATESCEDRVALTWNNLRKTLLVHQASEGLFDNDTGALLSLGREMFRLEILEDIARDK   420
A0A0H2USC0|IPA25_SHIFL    AADATESCEDRVALTWNNLRKTLLVHQASEGLFDNDTGALLSLGREMFRLEILEDIARDK   420
                          ************************************************************

A0A0H2USG1|IPA14_SHIFL    VRTLHFVDEIEVYLAFQTMLAEKLQLSTAVKEMRFYGVSGVTANDLRTAEAMVRSREENE   480
A0A0H2USC0|IPA25_SHIFL    VRTLHFVDEIEVYLAFQTMLAEKLQLSTAVKEMRFYGVSGVTANDLRTAEAMVRSREENE   480
                          ************************************************************

A0A0H2USG1|IPA14_SHIFL    FTDWFSLWGPWHAVLKRTEADRWAQAEEQKYEMLENEYSQRVADRLKASGLSGDADAERE   540
A0A0H2USC0|IPA25_SHIFL    FTDWFSLWGPWHAVLKRTEADRWAQAEEQKYEMLENEYSQRVADRLKASGLSGDADAERE   540
                          ************************************************************

A0A0H2USG1|IPA14_SHIFL    AGAQVMRETEQQIYRQLTDEVLALRLSENGSNHIA            575
A0A0H2USC0|IPA25_SHIFL    AGAQVMRETEQQIYRQLTDEVLA------------            563
                          ***********************
```

**Extended Data Fig. 1 | Sequence alignment of *Shigella flexneri* IpaH1.4/2.5.** The IpaH1.4 and IpaH2.5 protein sequences from *S. flexneri*, as listed in UNIPROT, were aligned using Clustal Omega. The structural domains are annotated below the sequences. IpaH1.4 and IpaH2.5 only differ in (1) the unstructured N-terminus, (2) the first alpha-helix preceding the leucine rich repeat, and (3) the C-terminal extension following the NEL domain. The leucine rich repeat, the NEL domain, and the interconnecting linker are 100% identical between the two proteins.

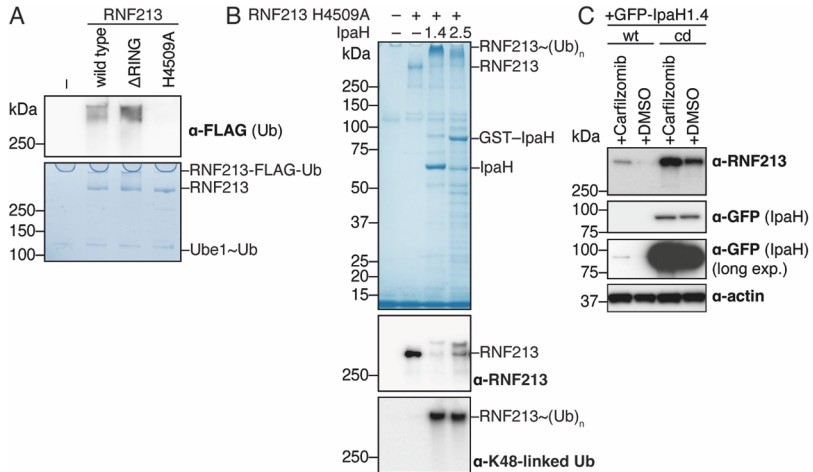

**Extended Data Fig. 2 | Choice of RNF213 mutant, lacking auto-ubiquitylation activity, for IpaH1.4/2.5-mediated ubiquitylation experiments. (a)** Auto-ubiquitylation assays of the indicated RNF213 variants incubated with Flag-tagged ubiquitin, UBE1 and UBCH5C (as described in the Methods). Upper: anti-Flag immunoblot. Lower: Coomassie stained SDS-PAGE. The results are representative of $n$ = 2 experiments. **(b)** *In vitro* ubiquitylation reaction containing Flag-tagged ubiquitin, UBE1, UBCH5C, the indicated IpaH proteins and enzymatically inactive RNF213 H4509A as substrate. Upper panel, Coomassie stained SDS-PAGE. Middle panel, immunoblot for RNF213. Bottom panel, immunoblot for K48-linked ubiquitin. The results are representative of $n$ = 3 experiments. **(c)** Immunoblot analysis of HeLa cell lysates expressing the indicated GFP-tagged IpaH proteins after treatment with proteasome inhibitor (Carfilzomib) or control (DMSO). Actin, loading control; cd, catalytically dead (E3 inactive) variant; wt, wild-type. The results are representative of $n$ = 3 experiments.

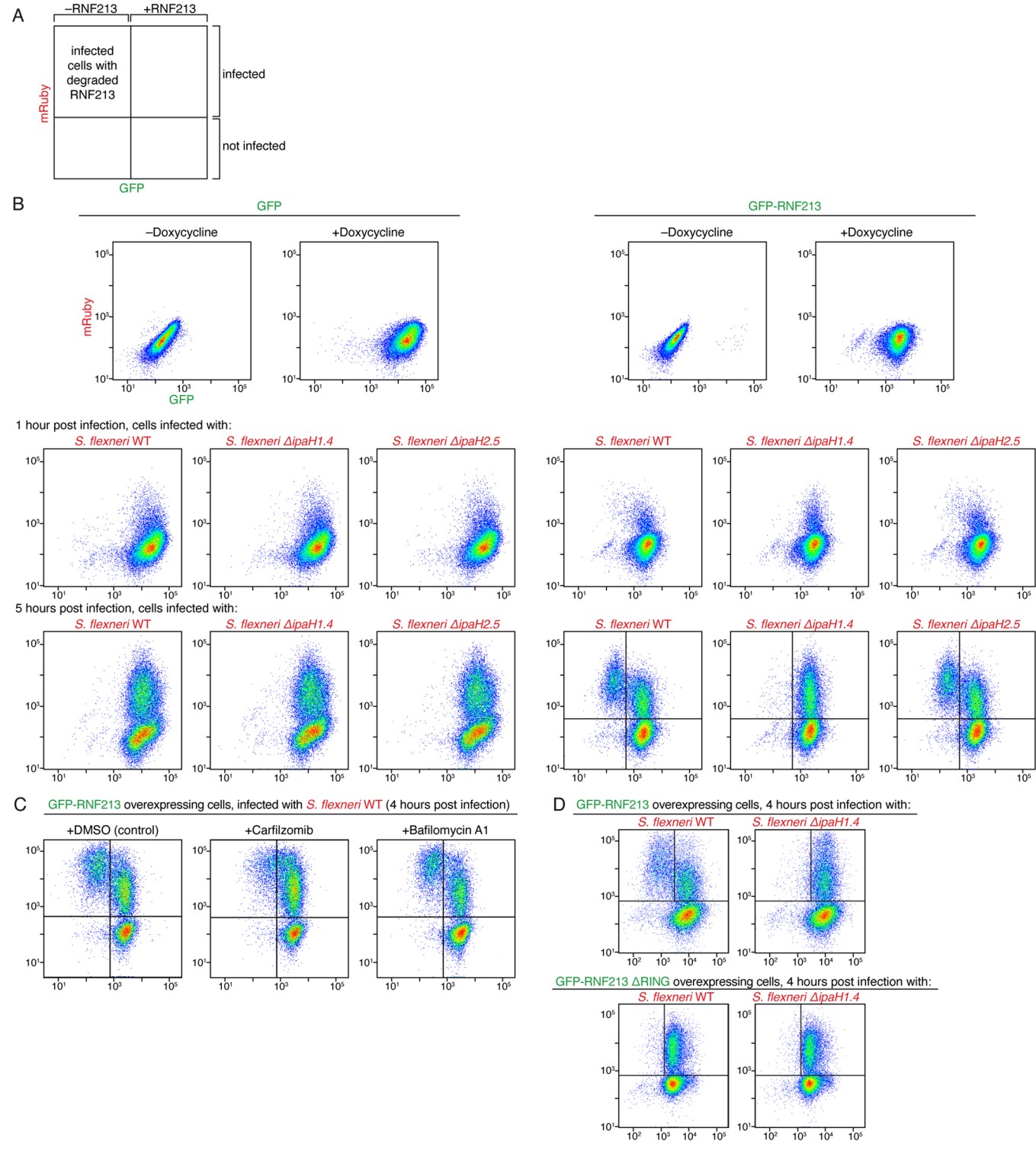

**Extended Data Fig. 3 | Quantification of *S. flexneri* induced degradation of RNF213 by flow cytometry.** (**a**) Cartoon representation of flow cytometry scatter plots below, indicating the identity of events observed in each quadrant. (**b**) Flow cytometry of GFP (left) and GFP-RNF213 (right) expressing MEFs induced with or without Doxycycline as indicated (top row); after induction with Doxycycline at 1 h (middle row) or 5 h post infection (bottom row) with the indicated strains of Ruby-labelled *S. flexneri*. (**c**) Flow cytometry of GFP-RNF213 expressing MEFs induced with Doxycycline and treated with the indicated inhibitors at 4 h post infection with Ruby-labelled *S. flexneri*. (**d**) Flow cytometry of GFP-RNF213 (upper) and GFP-RNF213 ΔRING (lower) expressing MEFs at 4 h post infection with the indicated strains of *S. flexneri*. All plots are on the same scale and include 30,000 events/plot.

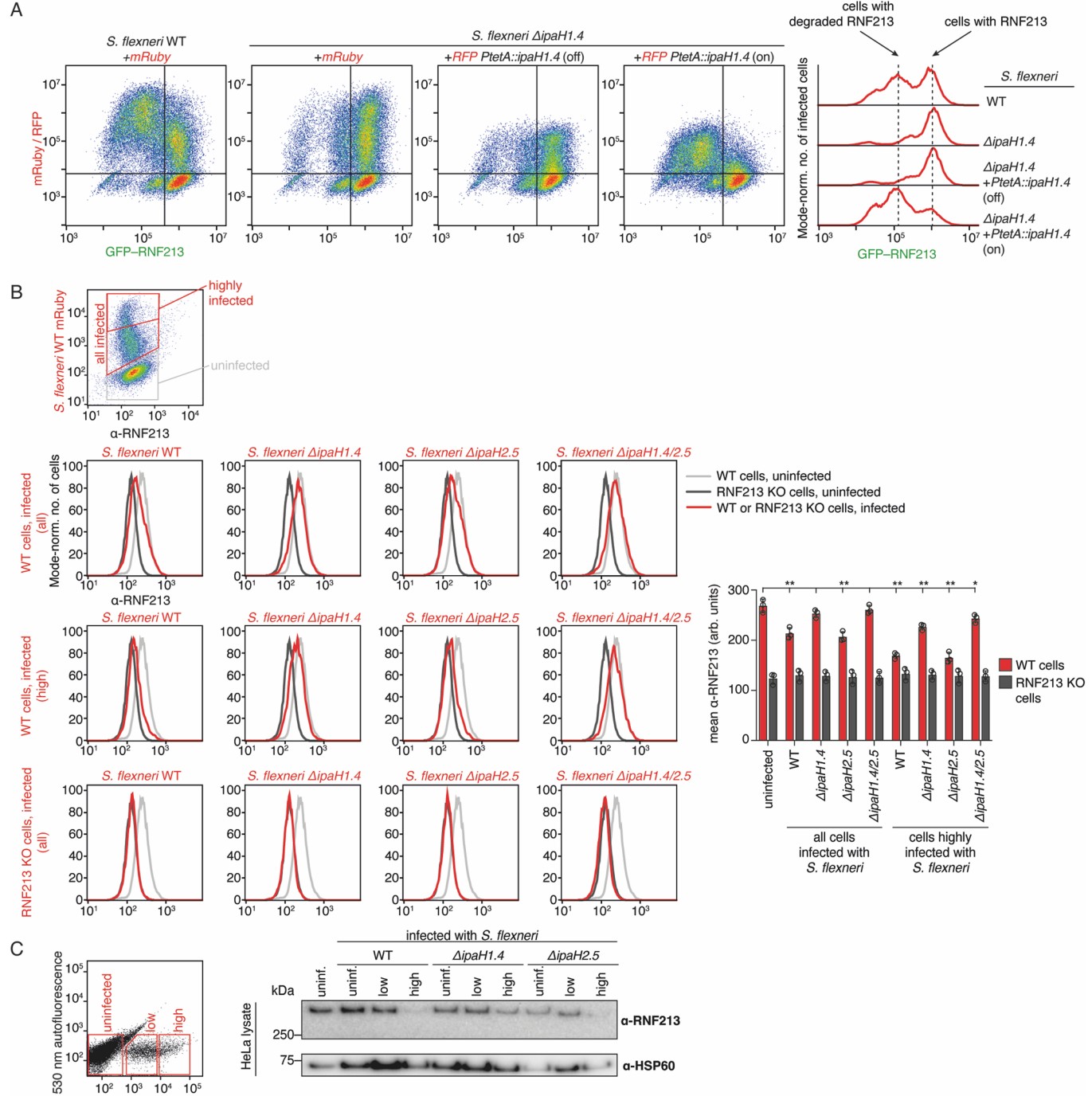

**Extended Data Fig. 4 | *Shigella flexneri* uses IpaH1.4 to degrade RNF213.**
(**a**) Complementation of *S. flexneri* Δ*ipaH1.4* with plasmid-encoded *ipaH1.4* rescues degradation of RNF213 in infected host cells. Left: Flow cytometry of MEFs expressing GFP-RNF213 under control of a Doxycycline-inducible promoter, induced with Doxycycline, and infected with the indicated strains of *S. flexneri* harboring plasmids encoding constitutively expressed mRuby or TagRFP-T and anhydrotetracycline-inducible *ipaH1.4*, induced with anhydrotetracycline (ATc) as indicated (off – not induced, on – induced). Cells were analyzed by flow cytometry at 5 h post infection. Right: histogram plots of GFP-RNF213 intensity in infected cells. (**b**) Analysis of endogenous RNF213 levels in *S. flexneri*-infected HeLa cells by flow cytometry. Wild type or RNF213 KO HeLa cells infected with the indicated strains of *S. flexneri* expressing mRuby were stained for endogenous RNF213. Levels of RNF213 are depicted as histograms

from the indicated cell populations, with bar heights in the accompanying bar graph denoting the mean and error bars denoting the standard deviations from $n = 3$ experiments; ** denotes the following p-values, in this order: p = 0.005, p = 0.003, p = 0.0003, p = 0.007, p = 0.0005, * denotes p = 0.048 (all derived from two-sided unpaired t-tests). The difference in staining between RNF213 WT and KO cells is indicative of the specific contribution of RNF213 to the assay. Infection with either *S. flexneri* WT or Δ*ipaH2.5*, but not Δ*ipaH1.4*, causes loss of RNF213 protein. (**c**) Left: Gating strategy for fluorescence-activated cell sorting (FACS) of HeLa cells. HeLa cells were infected with the indicated mRuby-expressing *S. flexneri* strains. At 4 h post infection the three indicated populations were isolated by FACS. Right: Immunoblot analysis of RNF213 in the indicated cell populations. HSP60 – loading control.

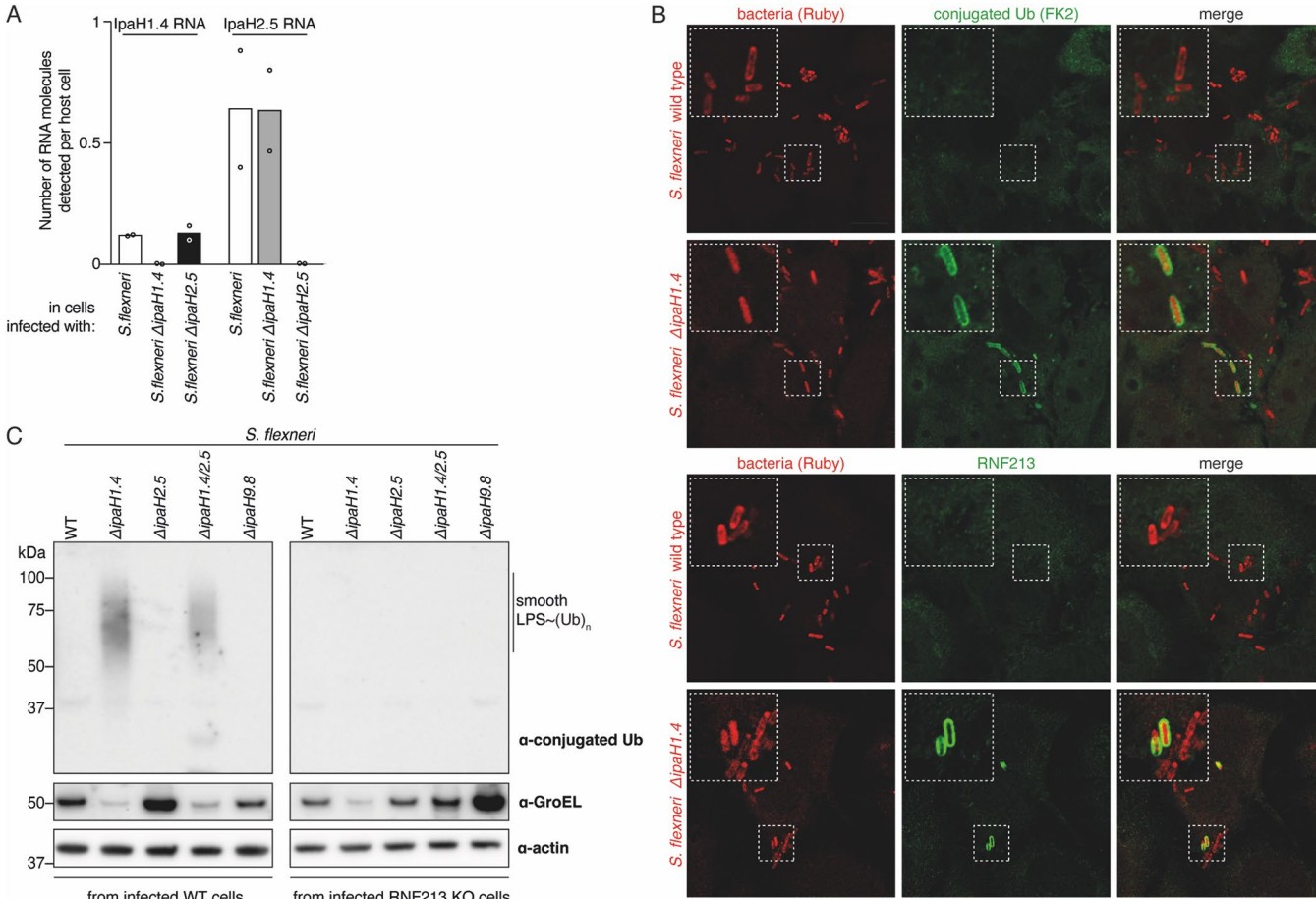

**Extended Data Fig. 5 | IpaH1.4 antagonizes RNF213-mediated LPS ubiquitylation in mouse and human cells.** (**a**) Detection of *ipaH1.4* and *ipaH2.5* transcripts in *Shigella flexneri*. The bars show mean values from *n* = 2 replicate experiments. The number of RNA molecules per host cell is estimated by dividing the number of RNA molecules (calculated relative to a plasmid for normalization, as described in the Methods) by the total number of cells used for each measurement ( ~ 10⁵). This represents a lower bound for the actual number of transcripts per bacterium due to the limited efficiency of the RNA extraction method. (**b**) Expanded view of Fig. 3b. Representative confocal micrographs of

HeLa cells at 4 h post infection with the indicated strains of Ruby-labeled *Shigella flexneri*, immunostained for conjugated ubiquitin (FK2) (left), or RNF213 (right). Dashed squares are magnified 2.5× in the insets. Scale bar, 10 μm. (**c**) Immunoblot analysis of the indicated strains of *S. flexneri* extracted from wild-type MEFs (left, same as Fig. 3d) and RNF213 KO MEFs (right) at 4 h post-infection. The LPS fractions were isolated by heat clearance of the bacterial lysates and probed for conjugated ubiquitin with FK2 antibody. The loading control (GroEL) was obtained from non-heat cleared bacterial lysates. The host cell loading control (actin) was obtained from clarified host cell lysates.

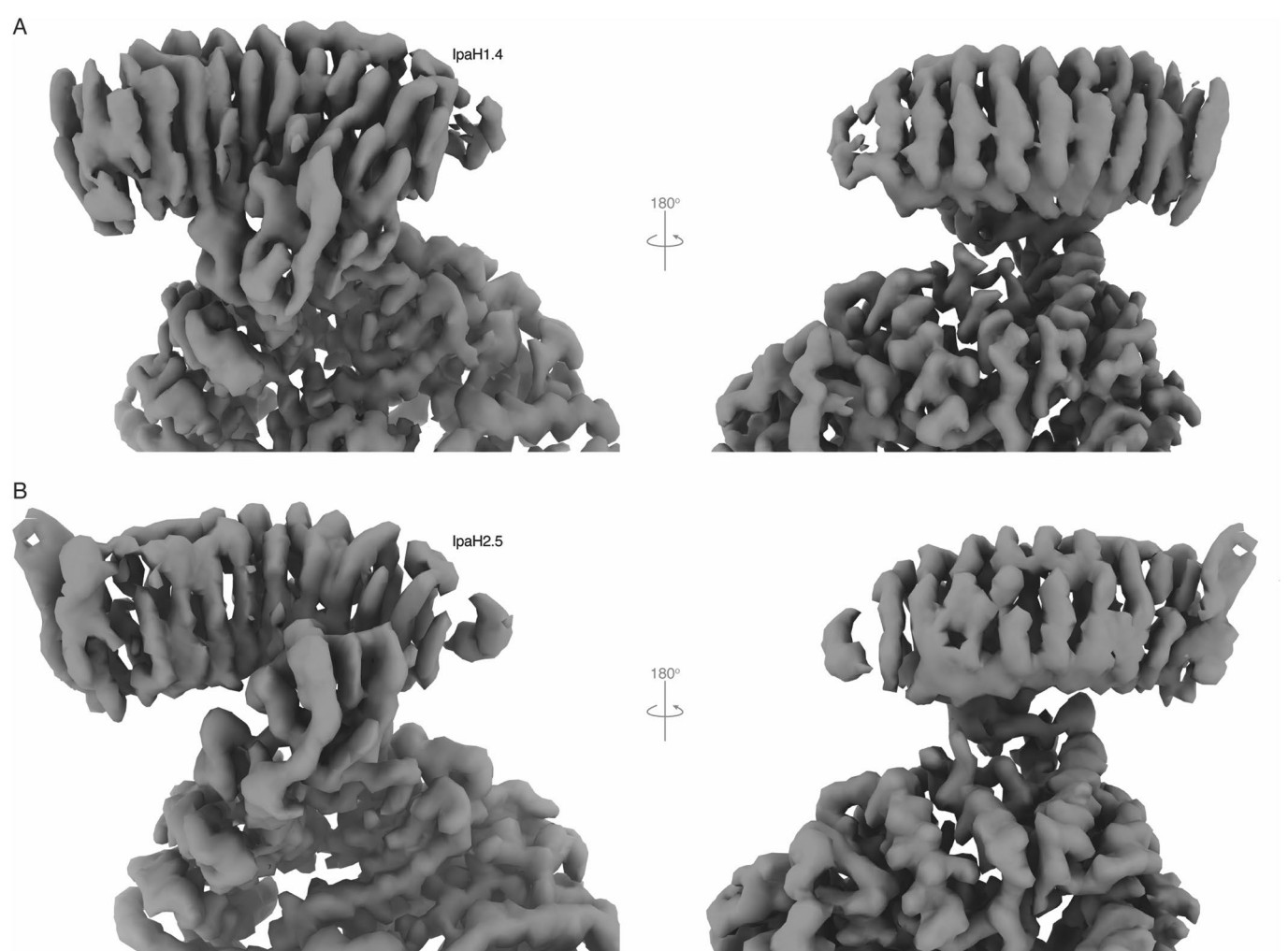

**Extended Data Fig. 6 | Structure of the IpaH2.5–RNF213 complex.** Comparison between the cryoEM maps of (**a**) IpaH1.4–RNF213 and (**b**) IpaH2.5–RNF213 complexes.

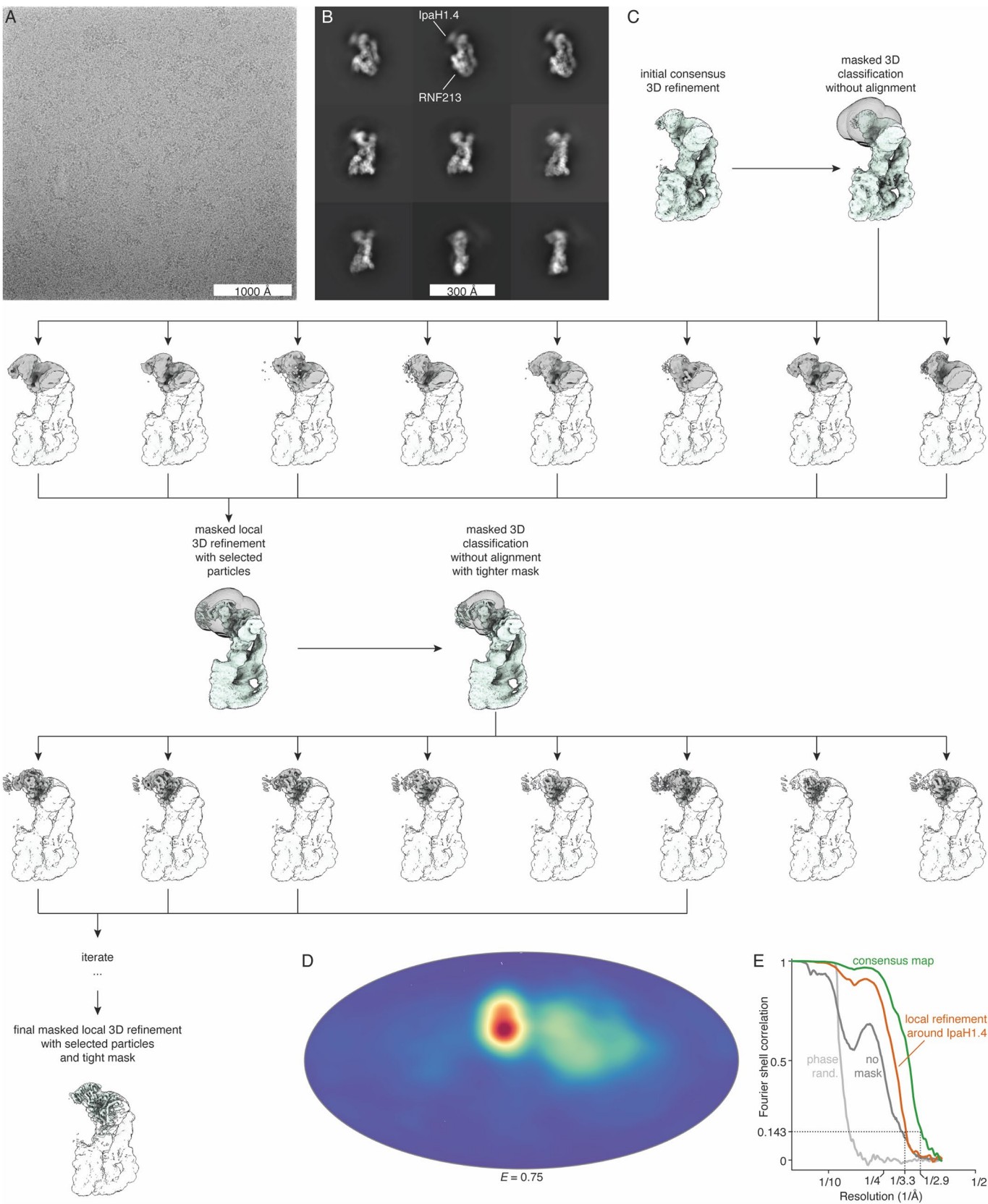

**Extended Data Fig. 7 | See next page for caption.**

**Extended Data Fig. 7 | CryoEM data processing.** (**a**) A representative micrograph, acquired at 1.5 µm defocus, from the IpaH1.4–RNF213 dataset (comprising 16,931 micrographs in total), low-pass filtered to 10 Å for display. (**b**) Representative 2D class averages of the IpaH1.4–RNF213 complex. (**c**) The flow chart summarizes the three-dimensional processing strategy, aimed at removing RNF213 particles without IpaH1.4 bound and at improving the resolution around the flexible region of the IpaH1.4 LRR and RNF213 RING. All processing steps after the initial consensus 3D refinement were performed in RELION.

(**d**) Orientation distribution of the particles used in the final reconstruction of the IpaH1.4–RNF213 complex, plotted on a Mollweide projection. The efficiency of the orientation distribution, $E$, is 0.75. (**e**) Fourier shell correlation as function of resolution is plotted for the best consensus masked map (green), the best masked map for the locally refined region of the IpaH1.4 LRR and RNF213 RING (orange), the unmasked consensus map (dark grey), and the phase-randomized consensus map (light grey).

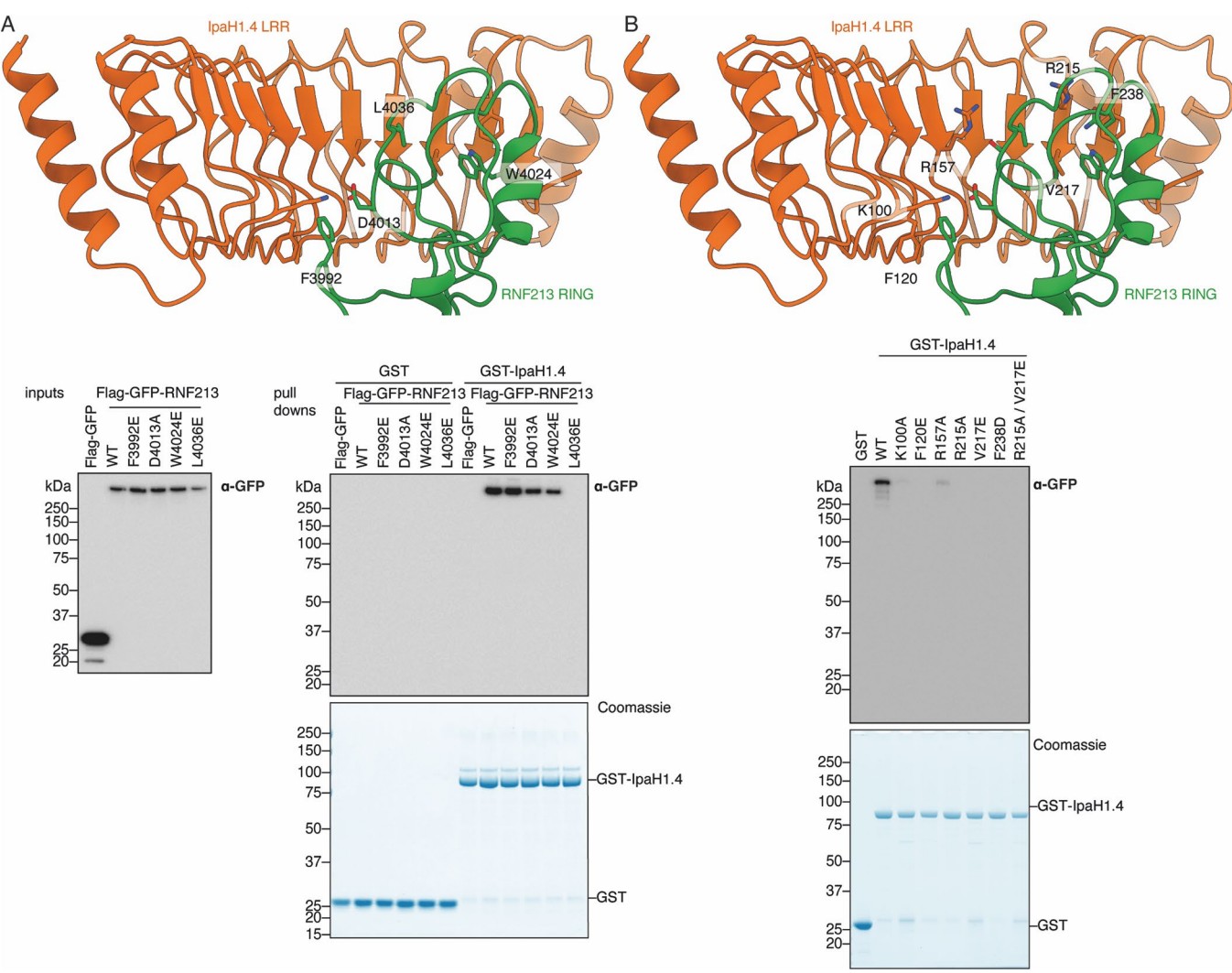

**Extended Data Fig. 8 | *In vitro* validation of the IpaH1.4–RNF213 binding interface.** (**a**) Pulldown of the indicated Flag-GFP-tagged RNF213 proteins from cell lysates upon incubation with purified GST-IpaH1.4 (or GST as negative control) as bait. Mutated RNF213 residues and their interaction partners in IpaH1.4 are highlighted in the cartoon. (**b**) Pulldown of Flag-GFP-RNF213 from cell lysate upon incubation with GST or GST-tagged IpaH (wild type or mutant) as bait. Mutated residues in IpaH1.4 and their interaction partners in RNF213 are highlighted in the cartoon. The results in both panels are representative of $n = 3$ experiments each.

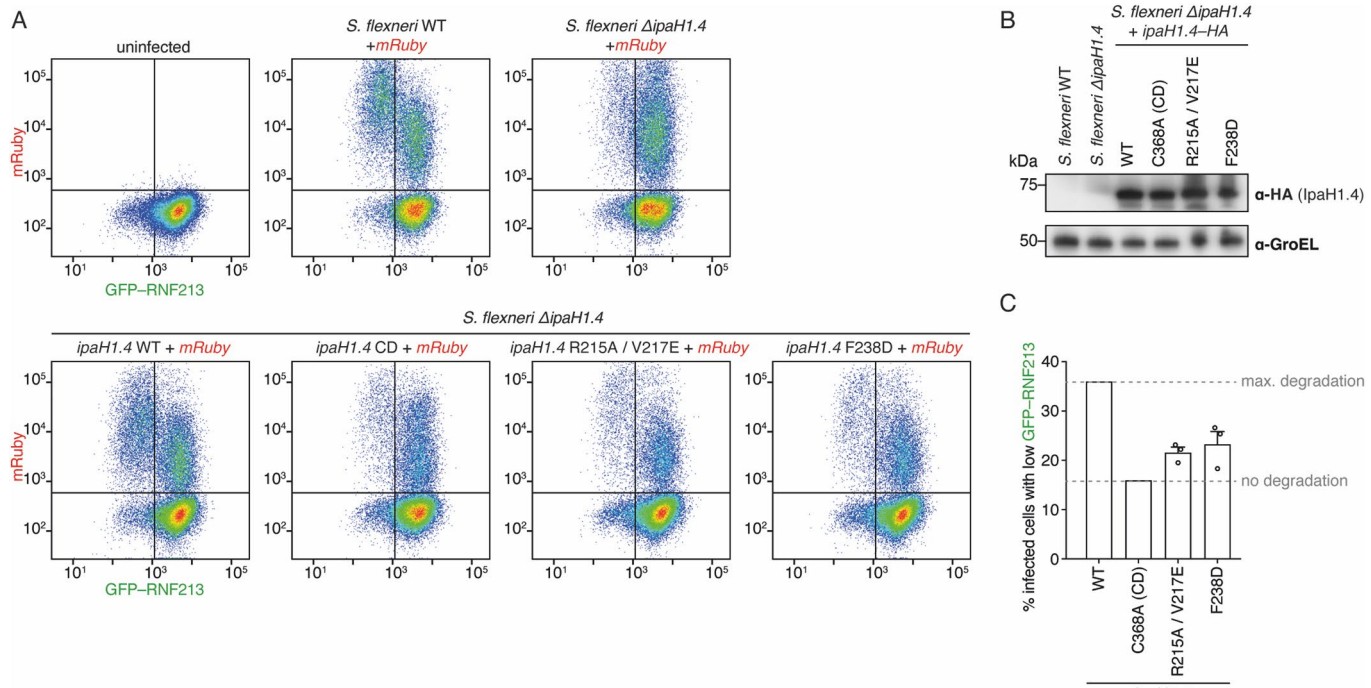

**Extended Data Fig. 9 | Validation of the IpaH1.4–RNF213 binding interface in a cell-based infection assay. (a)** Flow cytometry of MEFs expressing doxycycline-induced GFP-RNF213 and infected with the indicated strains of *S. flexneri* that harbor plasmids encoding either constitutively expressed mRuby alone or bicistronic C-terminally HA-tagged IpaH1.4 alleles together with mRuby. Cells were analyzed by flow cytometry at 5 h post infection. A total of 50,000 events were collected for each sample. Upper left quadrant depicts infected cells in which GFP-RNF213 levels are low, and upper right – those in which levels are high. **(b)** Immunoblot analysis of the indicated strains of *S. flexneri* to assess expression of HA-tagged IpaH1.4 upon complementation. CD, catalytically dead

(C368A); WT, wild-type; GroEL, loading control. The results are representative of *n* = 2 experiments. **(c)** Quantification of the fraction of infected cells in which the intensity of GFP-RNF213 is low. The dashed lines indicate the baselines of no and maximal RNF213 degradation, determined by the fraction of cells in this population when infected with *S. flexneri ΔipaH1.4* complemented with catalytically inactive or wild-type IpaH1.4, respectively. CD, catalytically dead (C368A); WT, wild-type. Data are the average of duplicate wells from a single experiment, with error bars representing mean ± standard deviation of three independent clones complemented with IpaH1.4 R215A/V217E or F238D.

# Reporting Summary

## Statistics

For all statistical analyses, confirm that the following items are present in the figure legend, table legend, main text, or Methods section.

| n/a | Confirmed | |
|---|---|---|
| ☐ | ☒ | The exact sample size (*n*) for each experimental group/condition, given as a discrete number and unit of measurement |
| ☐ | ☒ | A statement on whether measurements were taken from distinct samples or whether the same sample was measured repeatedly |
| ☐ | ☒ | The statistical test(s) used AND whether they are one- or two-sided<br>*Only common tests should be described solely by name; describe more complex techniques in the Methods section.* |
| ☒ | ☐ | A description of all covariates tested |
| ☒ | ☐ | A description of any assumptions or corrections, such as tests of normality and adjustment for multiple comparisons |
| ☐ | ☒ | A full description of the statistical parameters including central tendency (e.g. means) or other basic estimates (e.g. regression coefficient) AND variation (e.g. standard deviation) or associated estimates of uncertainty (e.g. confidence intervals) |
| ☐ | ☒ | For null hypothesis testing, the test statistic (e.g. *F*, *t*, *r*) with confidence intervals, effect sizes, degrees of freedom and *P* value noted<br>*Give P values as exact values whenever suitable.* |
| ☒ | ☐ | For Bayesian analysis, information on the choice of priors and Markov chain Monte Carlo settings |
| ☒ | ☐ | For hierarchical and complex designs, identification of the appropriate level for tests and full reporting of outcomes |
| ☒ | ☐ | Estimates of effect sizes (e.g. Cohen's *d*, Pearson's *r*), indicating how they were calculated |

*Our web collection on statistics for biologists contains articles on many of the points above.*

## Software and code

Policy information about availability of computer code

| Data collection | Commercial software: EPU v3.7 (cryoEM data collection); Digital Micrograph v3; Zeiss ZEN (fluorescence microscopy imaging); BioRad Image Lab Touch v2.4 (imaging of blots and gels); UNICORN v5.0 (AKTA for protein purification); BD FACS Diva v9.0 (flow cytometry). No custom software. |
|---|---|
| Data analysis | Open source and commercial software: RELION v5.0, CTFFIND v4.1, Resmap v1.1, cryoEF v1.2, pyEM v0.4, cryoSPARC v4.2-4.3 (cryoEM data analysis); ChimeraX v1.7-1.8, ISOLDE v1.6, coot v 0.9, Phenix v1.17 (atmoic model building, refinement, and visualization); FIJI (fluorescence image display); GraphPad Prism v10, Anaconda v3 for Python v3.6 (graphs); FlowJo v9-10 (flow cytometry). No custom software. |

For manuscripts utilizing custom algorithms or software that are central to the research but not yet described in published literature, software must be made available to editors and reviewers. We strongly encourage code deposition in a community repository (e.g. GitHub). See the Nature Portfolio guidelines for submitting code & software for further information.

## Data

Policy information about availability of data

All manuscripts must include a data availability statement. This statement should provide the following information, where applicable:
- Accession codes, unique identifiers, or web links for publicly available datasets
- A description of any restrictions on data availability
- For clinical datasets or third party data, please ensure that the statement adheres to our policy

The cryoEM maps and the refined atomic models have been deposited in Electron Microscopy Data Bank and the Protein Data Bank under the accession codes EMD 50913–50929 and PDB 9G08/9G09, respectively. Other data are available within the article and its associated supplementary information files. The following publicly available atomic models were retrieved from the Protein Data Bank for the purpose of model building or comparison: PDB 7V8G, 7V8F, 7YA7, 7YA8, 8S24.

## Research involving human participants, their data, or biological material

Policy information about studies with human participants or human data. See also policy information about sex, gender (identity/presentation), and sexual orientation and race, ethnicity and racism.

| | |
|---|---|
| Reporting on sex and gender | No human participants involved in the research. |
| Reporting on race, ethnicity, or other socially relevant groupings | No human participants involved in the research. |
| Population characteristics | No human participants involved in the research. |
| Recruitment | No human participants involved in the research. |
| Ethics oversight | No human participants involved in the research. |

Note that full information on the approval of the study protocol must also be provided in the manuscript.

# Field-specific reporting

Please select the one below that is the best fit for your research. If you are not sure, read the appropriate sections before making your selection.

☒ Life sciences          ☐ Behavioural & social sciences          ☐ Ecological, evolutionary & environmental sciences

For a reference copy of the document with all sections, see nature.com/documents/nr-reporting-summary-flat.pdf

# Life sciences study design

All studies must disclose on these points even when the disclosure is negative.

| | |
|---|---|
| Sample size | No statistical methods were used to determine sample size. For cryoEM, the sample size was determined by the data collection time available and was sufficient as indicated by the resolution reached in the refined maps. The numbers of micrographs and particles are listed in Supplementary Table S1. For blots, the sample size (number of cells per sample) was chosen so as to give unambiguous signal in positive control and loading control lanes, and was always greater than 10,000 cells per lane. For flow cytometry and cell counting, the sample sizes are similar to those generally used in the field and are stated in the figure legends; the statistical results from biological replicates show that the sample sizes chosen were sufficient for statistical significance and reproducibility. For flow cytometry, analysing at least 10,000 cells per sample is sufficient to ensure the error on the Poisson event counting is less than or equal to 1%, which is smaller than the typical differences between the analysed samples and the standard deviation between biological replicates. For cell counting the sample size was also sufficient to ensure that the counting error is less than the standard deviation between biological replicates, hence further increase in the sample size was not deemed necessary. Similar sample sizes for counting of infected cells have been previously used by us and others, see e.g. Otten et al, Nature, 2021. |
| Data exclusions | No technical replicates were excluded. For cryoEM data processing, particles were excluded using standard classification algorithms and those excluded corresponded to contaminants, false picks, or incomplete complexes (as described in Extended Data Figure 7C). |
| Replication | Experiments shown in the manuscript were reliably reproducible by multiple authors. Number of independent biological repeats are stated in the figure legends. |
| Randomization | Cells and bacteria from common pools were split randomly for the different experiments. For the cryoEM data analysis, the particles were split into random subsets (mini-batches) or half-sets during 2D/3D classification and refinement. |
| Blinding | Blinding was not relevant to this study as the experiments were conducted by individual researchers and required keeping note of the identity of the specimens. |

# Reporting for specific materials, systems and methods

We require information from authors about some types of materials, experimental systems and methods used in many studies. Here, indicate whether each material, system or method listed is relevant to your study. If you are not sure if a list item applies to your research, read the appropriate section before selecting a response.

## Materials & experimental systems

| n/a | Involved in the study |
|---|---|
| ☐ | ☒ Antibodies |
| ☐ | ☒ Eukaryotic cell lines |
| ☒ | ☐ Palaeontology and archaeology |
| ☒ | ☐ Animals and other organisms |
| ☒ | ☐ Clinical data |
| ☒ | ☐ Dual use research of concern |
| ☒ | ☐ Plants |

## Methods

| n/a | Involved in the study |
|---|---|
| ☒ | ☐ ChIP-seq |
| ☐ | ☒ Flow cytometry |
| ☒ | ☐ MRI-based neuroimaging |

## Antibodies

| | |
|---|---|
| Antibodies used | FK2 (Enzo Life Science, BML-PW8810, against conjugated ubiquitin, 1:1000), anti-GroEL (Enzo Life Science, ADI-SPS-875-F, 1:2000), anti-actin (Abcam, ab8227, 1:1000), anti-RNF213 (Merck, HPA003347 and HPA026790, 1:1000), anti-Flag–M2–HRP (Merck, A8592, 1:1000), anti-GFP (JL8, Clontech, 632381, 1:2000), anti-K48-linked ubiquitin (Abcam, ab140601, 1:1000), anti-Salmonella typhimurium LPS (BioRad, 8210-0407, 1:100), Salmonella O Antisera Group B (BD Difco, 229481, 1:100). Secondary antibodies: AF488-Goat-anti-Rabbit IgG (Invitrogen A11034, 1:500), Goat Anti-Mouse Immunoglobulins/HRP (Dako P0447, 1:5000), Goat Anti-Rabbit Immunoglobulins/HRP (Dako P0448, 1:5000). |
| Validation | Antibodies were validated by the manufacturers by western blotting as follows: FK2 - on ubiquitin chains (K29, K48, K63-linked); anti-GroEL - on recombinant E. coli GroEL; anti-actin - on HeLa cell lysate; anti-RNF213 - on A549 and HEK293 cell lysate; anti-Flag-M2-HRP - on Flag-transfected COS cells; anti-GFP - on cell lysate from HEK293 stably expressing AcGFP1; anti-K48-linked ubiquitin - on K48-linked-Ub2-7 recombinant protein. Anti-Salmonella typhimurium LPS antibody was validated by the manufacturerusing ELISA and IF against Salmonella typhimurium and negative controls (S. paratyphi A, S. choleraesuis, S. newport, S. enteriditis, S. anatum, S. selandia, E. coli 055:B5, E. coli K12, Klebsiella pneumoniae). Salmonella O Antisera Group B was validated by the manufacturer using the slide agglutination test on a variety of Salmonella cultures. Secondary antibodies were validated by the manufacturer using ELISA with immunoglobulins from various species. |

## Eukaryotic cell lines

Policy information about cell lines and Sex and Gender in Research

| | |
|---|---|
| Cell line source(s) | HeLa cells, HEK293T cells and mouse embryonic fibroblasts were obtained from ATCC. The RNF213 KO cell lines have been previously described (Otten et al, Nature, 2021). |
| Authentication | No authentication was performed. |
| Mycoplasma contamination | All cells tested negative for mycoplasma. |
| Commonly misidentified lines (See ICLAC register) | No commonly misidentified cell line was used. |

## Plants

| | |
|---|---|
| Seed stocks | No plants involved in the research. |
| Novel plant genotypes | No plants involved in the research. |
| Authentication | No plants involved in the research. |

# Flow Cytometry

## Plots

Confirm that:

☒ The axis labels state the marker and fluorochrome used (e.g. CD4-FITC).

☒ The axis scales are clearly visible. Include numbers along axes only for bottom left plot of group (a 'group' is an analysis of identical markers).

☒ All plots are contour plots with outliers or pseudocolor plots.

☒ A numerical value for number of cells or percentage (with statistics) is provided.

## Methodology

| | |
|---|---|
| Sample preparation | Cultured cells, live or fixed in 4% paraformaldehyde, and washed in PBS, as described in the Mehtods section. |
| Instrument | BD LSRFortessa, BD FACS AriaFusion |
| Software | Data collection: BD FACS Diva v 9.0. Data analysis: FlowJo v9-10. |
| Cell population abundance | A total of 10,000 – 30,000 cells per sample were analyzed, as specified in the text. The relevant cell population abundances are quantified in Figures 3A and 4G, as described in Extended Data Figure 3A. The relevant cell population abundances are quantified in Extended Data Figures 4 and 9, as described therein. |
| Gating strategy | No FSC/SSC gating was applied. Fluorescence +/- boundaries for Figure 3 were defined as shown in Extended Data Figure 3A, based  on the controls shown in the upper row of Extended Data Figure 3B. For the experiments shown in Extended Data Figures 4 and 9, the gates are shown for each panel separately. |

☒ Tick this box to confirm that a figure exemplifying the gating strategy is provided in the Supplementary Information.

