## [Peer Review File · Nature Structural & Molecular Biology]

Shigella flexneri evades LPS ubiquitylation through IpaH1.4-mediated degradation of RNF213

Corresponding Author: Dr Felix Randow

Version 0:

Decision Letter:

13th Aug 2024

Dear Dr. Randow,

Thank you again for submitting your manuscript "Shigella flexneri evades LPS ubiquitylation through IpaH1.4-mediated degradation of RNF213". I apologize for the delay in responding, which resulted from the difficulty in obtaining suitable referee reports. Nevertheless, we now have comments (below) from the 3 reviewers who evaluated your paper. In light of those reports, we remain interested in your study and would like to see your response to the comments of the referees, in the form of a revised manuscript.

You will see that while reviewers appreciate the results, they raise several concerns which will need to be addressed in a revision. Specifically, in line with referee #1 and #2 comments, please provide western blot analysis to complement the FACS data on the degradation of RNF213. Please also validate the structural results using mutagenesis, as requested by referees #1 and #3. Moreover, please provide missing controls, quantification and statistical analysis as requested by the referees. We also agree with referee #2 that investigating the requirements for IpaH1.4 but not IpaH2.5 further, will strengthen the manuscript, as well as the proposed experiments testing the requirement of LUBAC.

Please be sure to address/respond to all concerns of the referees in full in a point-by-point response and highlight all changes in the revised manuscript text file. If you have comments that are intended for editors only, please include those in a separate cover letter.

We expect to see your revised manuscript within 6 weeks. If you cannot send it within this time, please contact us to discuss an extension; we would still consider your revision, provided that no similar work has been accepted for publication at NSMB or published elsewhere.

Reporting Summary:

Please note that all key data shown in the main figures as cropped gels or blots should be presented in uncropped form, with molecular weight markers. These data can be aggregated into a single supplementary figure item. While these data can be displayed in a relatively informal style, they must refer back to the relevant figures. These data should be submitted with the final revision, as source data, prior to acceptance, but you may want to start putting it together at this point.

Data availability: this journal strongly supports public availability of data. All data used in accepted papers should be available via a public data repository, or alternatively, as Supplementary Information. If data can only be shared on request, please explain why in your Data Availability Statement, and also in the correspondence with your editor. Please note that for some data types, deposition in a public repository is mandatory - more information on our data deposition policies and available repositories can be found below:

<https://www.nature.com/nature-research/editorial-policies/reporting-standards#availability-of-data>

Link Redacted

Sincerely,

Katarzyna Ciazynska, PhD
(she/her)
Associate Editor
Nature Structural & Molecular Biology
<https://orcid.org/0000-0002-9899-2428>

Referee expertise:

Referee #1: structural biology, microbiology, immunology

Referee #2: microbiology

Referee #3: structural biology, chemical biology

Reviewers' Comments:

Reviewer #1:

Remarks to the Author:

This manuscript evaluates the quandary of how cytosolic pathogens can survive in spite of the recently discovered RNF213 defense system of lipid ubiquitination. RNF213 is a host E3 ligase that targets various pathogens for clearance through its ubiquitination activity. Naydenova et al focus on the pathogen *Shigella flexneri* and discover that the secreted effector IpaH1.4 binds to the RING domain of RNF213, resulting in ubiquitination and degradation. A series of cryoEM structures visualize this interaction and identify an interesting feature of IpaH1.4 that allows it to target multiple members of host defense. The manuscript is very nicely presented and well-written. The experiments and data are technically sound and nicely organized. Some conclusions could be strengthened with consideration of the concerns detailed below. Barring that, this work will be of high interest to the ubiquitin and host-pathogen interaction communities.

Major concerns:

1. The authors present flow cytometry data in support of IpaH1.4-induced proteasomal degradation of RNF213 in the context of *Shigella* infection. A simplified western blot, perhaps of cells expressing IpaH1.4 and RNF213 in the absence of infection, would be important to show that proteasomal inhibition results in an accumulation of ubiquitylated RNF213. This evidence would fit nicely into the flow of the results following Figure 3A.
2. What is the timing of this RNF213 antagonism? Is IpaH1.4 secreted and active against RNF213 by the time that *Shigella* are exposed to the cytosol following vacuolar rupture?
3. The experiment presented in Figure 3C should be expanded to also visualize the status of RNF213 as well as a host protein loading control.
4. The authors present a nice structure that visualizes the IpaH1.4:RNF213 interaction but do not validate the IpaH1.4 interface with subsequent experiments. Revisiting the experiment presented in Figure 3C with a Δ IpaH1.4 strain complemented with WT or mutated IpaH1.4 would a) show that the LPS ubiquitylation can be blocked with WT complementation, and b) validate the binding interface.

Minor concerns:

1. It is unclear from figures, figure legends, text, and methods how LPS was immunostained for western blotting.
2. The authors propose that IpaH ligases induce proteasomal degradation of their targets by catalyzing K48 ubiquitination. Could the authors show this for RNF213? Perhaps by performing a western blot on the in vitro reaction with a K48-specific antibody?
3. In the introduction, the authors are missing a reference to Hansel et al., Cell 2021 for the IpaH7.8-mediated degradation of gasdermin.
4. In the introduction, the authors state that "in all known cases, the IpaH ligase targets important anti-bacterial host proteins for proteasomal degradation through conjugation of K48-linked ubiquitin chains", yet the IpaH9.8 study on NEMO degradation suggested K27-linked chains.
5. In Figure 2C, would it be possible to also blot for RNF213 as the substrate? The lower anti-FLAG blot should be labeled with MW markers. It should also be specified which form of UbcH5 is used (the methods state UbcH5c).
6. What statistical test was used for Figure 3A?
7. What is the difference between Figure 3B and Extended Data Figure 5?
8. Given that the images presented in Figure 3B and from HeLa cells and the quantified data shown in the bar plots are from MEFs, the authors should consider splitting them into separate data panels.
9. At the bottom of page 9, I believe the authors may want to also cite Extended Data Figure S3D alongside Figure 4G.
10. In the Discussion, the sentence "The E3 ligase activity of RNF213 and IpaH-family effectors are carried" may be missing the word "out".
11. In the Discussion section, the authors should consider citing Pruneda et al., Mol Cell 2012 as part of the discussion on the Arg/Lys 'linchpin'.
12. In the first sentence under the Bacterial Infections section of the Methods, there is a typo following 3.5.
13. In the Methods section, details on the buffer systems used for the Resource Q column were not provided.

Reviewer #2:

Remarks to the Author:

How *Shigella* may prevent LPS ubiquitination has been of great interest since the transformative discovery of LPS ubiquitination using *Salmonella* (Nature 2021). In this report, the authors discover that cytosol-dwelling *Shigella* avoids cell-autonomous immunity by secreting the bacterial effector IpaH1.4, which ubiquitinates the E3 ligase RNF213 for proteasomal degradation. Authors resolve the structure of RNF213 bound to IpaH1.4/2.5, providing high level of detail. Cellular and biochemical methods are performed (infection assays, pull-downs, protein purifications, in vitro ubiquitination, cryoEM), providing in depth mechanistic and meaningful insights on RNF213 biology. Although the role of RNF213 in *Shigella* infection control is not yet clear, this manuscript will excite researchers in the field of *Shigella* and cell-autonomous immunity. Many future research avenues emerge from this high-quality work, and I offer comments for the authors to consider for this report or for future work.

1. To investigate degradation of RNF213, authors present a FACS based assay where they measure degradation of GFP-RNF213 in WT *Shigella* and mutants (Δ ipaH1.4, Δ ipaH2.5). It would be complementary if authors show this effect is also produced with the endogenous RNF213 measured by western blot in sorted infected cells.

2. While WT *Shigella* and Δ ipaH1.4 replicate similarly during infection (Fig 3D), it seems that both strains do worse in the absence of RNF213? Could the authors explain this effect or offer any hypothesis? Also, data plotted coming from N=1 biological replicate?

3. IpaH1.4, but not IpaH2.5, is required for RNF213 degradation by *Shigella*. These results are consistent with previous work studying LUBAC. The authors suggest a downstream mechanism could be lack of IpaH2.5 secretion during infection. Considering that expression of IpaH2.5 in HeLa cells is sufficient to prevent ubiquitination of *Salmonella* LPS (Fig 2B), these cells can be used to test this hypothesis.

4. Does RNF213 also ubiquitinate lipid A in *Shigella* LPS (similarly to *Salmonella*)? To test this, authors could construct a double mutant Δ rfaL Δ ipaH1.4.

5. Perform appropriate statistical tests for all microscopy quantifications (eg Fig 1C, Fig 3B).

7. The authors have previously shown that, at least in the case of cytosolic *Salmonella*, LPS ubiquitination by RNF213 is required for recruitment of LUBAC to the pathogen and further M1-linked ubiquitin chain formation. This has not been demonstrated in the case of *Shigella*. Why no LUBAC testing +/- RNF213 +/- IpaH1.4 in this study?

Minor points

1. The reference for the anti-LPS antibody used in Fig 1A is missing from the methods section.

2. Provide more detail on how the fold spread of *S. flexneri* was calculated in Fig 3E.

3. Fig 1A. % ubiquitination is not so black and white for *Shigella*? There is an extensive literature on WT *S. flexneri* associations with ubiquitin. Consistent with this, quantifications in Fig 3B is more in line with previous publications on *Shigella*-ubiquitin interactions.

4. Previous literature has suggested that *S. flexneri* IcsB is important to avoid autophagosome recruitment. Can authors include previous literature more directly to help contextualise their results with the rest of the field. For example, how would authors explain *S. flexneri* Δ icsB being recruited to autophagy in the presence of IpaH1.4?

5. Extended Fig 5 is the same as Fig 3B? Perhaps offering different images/examples would be more valuable to readers.

6. Crespillo-Casado et al 2024 is referenced throughout the text. I do not have access to this manuscript but hope it is available prior to publication of this work (and/or overlap is avoided).

Reviewer #3:

Remarks to the Author:

The manuscript reports the identification of IpaH1.4 as a *Shigella* E3 ligase effector protein that degrades the host E3 ligase RNF213. The lab recently showed that RNF213 is an important antimicrobial factor that targets LPS on *Salmonella* for ubiquitination, thereby signalling its degradation by xenophagy. Initial assays reveal that *Shigella* is not LPS ubiquitinated and identify IpaH1.4 and 2.5 as candidate RNF213 antagonists. Both bind and ubiquitylate RNF213, but only *Shigella* depleted of IpaH1.4 impairs GFP-RNF213 degradation. Finally, a cryoEM structure of a complex of the substrate targeting LRR region of IpaH bound to full-length RNF213 RING is solved. The LRR is contacting the RNF213 RING domain, as found with IpaH1.4 and the RING domain from HOIP. Overall, the paper is interesting but some aspects are underdeveloped and the catalytic E3 module from IpaH1.4 is not resolved. Moreover, the cellular proliferation of *Shigella* depleted of IpaH1.4 is unaffected. This raises an important question about the relevance of RNF213 recognition and degradation. For these reasons, and those below, it is difficult to fully support publication in NSMB.

Major points

RNF213 has already been solved so the novel structural insight is the inverted configuration of the RING domain compared to HOIP RING. As it can bind RING domains in two orientations, does this imply it is promiscuous? One would expect the identification of key residues that upon mutation impair and/or selectively impair binding. These should then be tested in cells for stabilization. The authors suggest other RING E3s might be targeted, which seems likely. An appreciation of the number would strengthen the manuscript. Insight could be generated with an affinity purification MS experiment using the LRR region as bait.

IpaH1.4 can target RNF213 and HOIP. In Figure 3C what happens if HOIP is knocked out?

It is proposed that IpaH2.5 depletion has no effect because it isn't expressed as highly as IpaH1.4. The authors should compare expression levels in infected cells to confirm if this is the case

In Figure 3B why was LPS immunostaining not carried out as with IpaH1.4?

There is no mxiE KO data in Figure 3B. Perhaps the labels are incorrect?

Data show that overexpressed GFP-RNF213 is degraded by Shigella. Can the authors also show that endogenous protein is degraded in infected cells?

Minor points

In Figure 1A colocalisation is restricted to specific subregions. Is there an explanation for this?

In the discussion the argument that, because IpaH targets the RING domain implies it must have an important function, is difficult to comprehend. Isn't the RING merely serving as a recruitment site to initiate degradation? If so, the site is irrelevant and this doesn't offer any insight into the normal role of the RING domain. Could the key role of IpaH1.4 in the context of RNF213 be to block the E3 activity of the RING domain by occluding E2 binding?

Figure legends would benefit from more information.

Version 1:

Decision Letter:

14th Nov 2024

Dear Dr. Randow,

Thank you again for submitting your manuscript "Shigella flexneri evades LPS ubiquitylation through IpaH1.4-mediated degradation of RNF213". I apologize for the delay in responding, which resulted from the difficulty in obtaining suitable referee reports. Nevertheless, we now have comments (below) from the 3 reviewers who evaluated your paper. In light of those reports, we remain interested in your study and would like to see your response to the comments of the referees, in the form of a revised manuscript.

You will see that reviewers 1 and 3 have expressed some remaining concerns. Specifically, reviewer #1 asks for further complementation experiments, to assuage their own concerns, as well as those of reviewer #3. Regarding reviewer #3 comments - please clearly acknowledge the caveats of the study, including addressing the comments about redundancy of function. Please be sure to address/respond to all concerns of the referees in full in a point-by-point response and highlight all changes in the revised manuscript text file. If you have comments that are intended for editors only, please include those in a separate cover letter.

We expect to see your revised manuscript within 6 weeks. If you cannot send it within this time, please contact us to discuss an extension; we would still consider your revision, provided that no similar work has been accepted for publication at NSMB or published elsewhere.

Reporting Summary:

- that unprocessed scans are clearly labelled and match the gels and western blots presented in figures.
- that control panels for gels and western blots are appropriately described as loading on sample processing controls

-- all images in the paper are checked for duplication of panels and for splicing of gel lanes.

Please note that all key data shown in the main figures as cropped gels or blots should be presented in uncropped form, with molecular weight markers. These data can be aggregated into a single supplementary figure item. While these data can be displayed in a relatively informal style, they must refer back to the relevant figures. These data should be submitted with the final revision, as source data, prior to acceptance, but you may want to start putting it together at this point.

Data availability: this journal strongly supports public availability of data. All data used in accepted papers should be available via a public data repository, or alternatively, as Supplementary Information. If data can only be shared on request, please explain why in your Data Availability Statement, and also in the correspondence with your editor. Please note that for some data types, deposition in a public repository is mandatory - more information on our data deposition policies and available repositories can be found below:

<https://www.nature.com/nature-research/editorial-policies/reporting-standards#availability-of-data>

Link Redacted

Sincerely,

Katarzyna Ciazynska, PhD
(she/her)
Senior Editor
Nature Structural & Molecular Biology
<https://orcid.org/0000-0002-9899-2428>

Reviewers' Comments:

Reviewer #1 (Remarks to the Author):

The authors have presented a nicely revised manuscript. Reviewer 3 raises a concern over the lack of a phenotype in either bacterial proliferation or spread following deletion of IpaH1.4. I am less concerned about this observation because, as the authors nicely explain, there are likely other effectors acting redundantly downstream of RNF213 to evade restriction. Both Reviewer 3 and I asked for additional validation of the LRR-RNF213 interface resolved in the cryoEM structure. In response, the authors have added a mutational analysis of both protein surfaces and identified residues that abrogate interactions observed by in vitro pulldowns. These data, presented in Extended Data Figure 8, could be explained better in the legend, as it isn't entirely clear which protein is being used as bait (though I believe it is GST-IpaH1.4 in both cases). Furthermore, it isn't clear why the loading of GST-IpaH1.4 is so different across the structural mutants. Were some of the mutants less stable? Lastly, both Reviewer 3 and I asked for some of these structural mutations to be introduced back into the cellular assay for RNF213 stability. The authors nicely show that the RING domain of RNF213 is required for IpaH1.4-mediated degradation, but what about more targeted mutations? I understand that it may be technically challenging to test multiple mutations in both proteins, but could the authors simply test one or two IpaH1.4 mutants in the complementation experiment presented in Extended Data Figure 4?

Outside of these points, I feel that the authors have satisfied all comments from both myself and Reviewer 3, and I continue to strongly believe that this is important work worthy of publication in NSMB.

Reviewer #2 (Remarks to the Author):

Thank you for addressing reviews and for clearly advancing the field.

Reviewer #3 (Remarks to the Author):

The authors have made a good effort to address my concerns. However, my primary issue was not discussed in the rebuttal which is that RNF213 has no antimicrobial activity towards *Shigella*, as evidenced from *Shigella* depleted of IpaH1.4 proliferating similarly to WT. I remain unconvinced that the characterisation of what might be a redundant function is appropriate for the readership of NSMB. My stance is exacerbated by the abstract and summary paragraph of the introduction, which are misleading because of the narrative that RNF213 LPS ubiquitination creates "eat me signals" that must destroy all gram negative bacteria by xenophagy. To support publication, I would expect more upfront transparency regarding potential redundancy in the infection model used, and if IpaH1.4 suppresses immune responses, the benefits to the pathogen remain unclear.

Version 2:

Decision Letter:

Our ref: NSMB-A49461B

9th Jan 2025

Dear Dr. Randow,

Thank you for submitting your revised manuscript "Shigella flexneri evades LPS ubiquitylation through IpaH1.4-mediated degradation of RNF213" (NSMB-A49461B). Please accept our apologies for the delay in this decision. The manuscript has now been seen by the original referees and their comments are below. The reviewers find that the paper has improved in revision, and therefore we'll be happy in principle to publish it in Nature Structural & Molecular Biology, pending minor revisions to satisfy the referees' final requests and to comply with our editorial and formatting guidelines.

To facilitate our work at this stage, it is important that we have a copy of the main text as a word file. If you could please send along a word version of this file as soon as possible, we would greatly appreciate it; please make sure to copy the NSMB account (cc'ed above).

Sincerely,
Kat

Katarzyna Ciazynska, PhD
(she/her)

Senior Editor
Nature Structural & Molecular Biology
<https://orcid.org/0000-0002-9899-2428>

Reviewer #1 (Remarks to the Author):

I commend the authors for addressing all reviewer comments, yielding a study that will be highly impactful across multiple fields. I disagree with the latest comments from Reviewer 3, and feel that the authors have presented their findings in a fair and accurate manner. As the authors state, the lack of a phenotype is not uncommon in this field and only adds to the excitement of what future work will find at this host-pathogen interface.

My only remaining suggestion is for the authors to check the new figure legend for Extended Data Figure 9, which mentions a GroEI loading control in panel C. In this case, I believe the authors intended to present the actin loading control that is part of panel B.

Version 3:

Decision Letter:

11th Mar 2025

Dear Dr. Randow,

We are now happy to accept your revised paper "Shigella flexneri evades LPS ubiquitylation through IpaH1.4-mediated degradation of RNF213" for publication as an Article in Nature Structural & Molecular Biology.

Your paper will be published online soon after we receive proof corrections and will appear in print in the next available issue. You can find out your date of online publication by contacting the production team shortly after sending your proof corrections.

Sincerely,

Katarzyna Ciazynska, PhD
(she/her)
Senior Editor
Nature Structural & Molecular Biology
<https://orcid.org/0000-0002-9899-2428>

Reviewer #1:

Remarks to the Author:

This manuscript evaluates the quandary of how cytosolic pathogens can survive in spite of the recently discovered RNF213 defense system of lipid ubiquitination. RNF213 is a host E3 ligase that targets various pathogens for clearance through its ubiquitination activity. Naydenova et al focus on the pathogen *Shigella flexneri* and discover that the secreted effector IpaH1.4 binds to the RING domain of RNF213, resulting in ubiquitination and degradation. A series of cryoEM structures visualize this interaction and identify an interesting feature of IpaH1.4 that allows it to target multiple members of host defense. The manuscript is very nicely presented and well-written. The experiments and data are technically sound and nicely organized. Some conclusions could be strengthened with consideration of the concerns detailed below. Barring that, this work will be of high interest to the ubiquitin and host-pathogen interaction communities.

We thank the reviewer for considering our work of ‘...high interest to the ubiquitin and host-pathogen communities’ and for their useful suggestions. We address all comments below.

Major concerns:

1. The authors present flow cytometry data in support of IpaH1.4-induced proteasomal degradation of RNF213 in the context of *Shigella* infection. A simplified western blot, perhaps of cells expressing IpaH1.4 and RNF213 in the absence of infection, would be important to show that proteasomal inhibition results in an accumulation of ubiquitylated RNF213. This evidence would fit nicely into the flow of the results following Figure 3A.

We have addressed the reviewer’s request experimentally. As predicted by the reviewer, new data (**Extended Data Figure 2C**) reveal that proteasome inhibition indeed leads to accumulation of RNF213.

2. What is the timing of this RNF213 antagonism? Is IpaH1.4 secreted and active against RNF213 by the time that *Shigella* are exposed to the cytosol following vacuolar rupture?

We thank the reviewer for this interesting question. Unfortunately, we lack the necessary tools to investigate this question directly as no antibody against IpaH1.4 is available to us. We also cannot address the question indirectly, i.e. by determining the time point of functional interference of IpaH1.4 with RNF213 using live microscopy, as this experiment would require the expression of fluorescently tagged RNF213 at levels unlikely to match endogenous RNF213, thus giving artificial readings.

However, in the interim, insights into the general timing of the IpaH antagonism can be gained from the early works of the Sasakawa group (Toyotome et al, JBC, 2001; Ashida et al, Molecular Microbiology, 2006). They have shown that IpaH effectors are delivered into cells only after *Shigella* invasion has taken place, i.e. not from extracellular *Shigella* (unlike IpaB/C/D which are delivered prior to invasion). Thus, IpaH effectors belong to the ‘second wave’ of secreted effectors. We therefore expect that secretion of IpaH1.4 starts when *Shigella* invades the host cell but before it breaks

the vacuole, thus enabling IpaH1.4 to antagonize RNF213 before Shigella enters the cytosol and becomes exposed to the action of RNF213.

3. The experiment presented in Figure 3C should be expanded to also visualize the status of RNF213 as well as a host protein loading control.

We thank the reviewer for this suggestion.

Regarding the host protein loading control, an actin blot has been added to the panel (which has been relabeled as **Figure 3D**).

Regarding the request to monitor RNF213 levels, since no antibody against mouse RNF213 is commercially available, we tested multiple anti-human RNF213 antibodies for cross-reactivity against the murine protein. Unfortunately, no suitable antibody was identified. However, new information closely related to the reviewer's request was added to the new **Extended Data Figure 4C**, where we demonstrate the effect of IpaH proteins delivered by Shigella on the level of endogenous human RNF213, as assessed by Western blotting.

4. The authors present a nice structure that visualizes the IpaH1.4:RNF213 interaction but do not validate the IpaH1.4 interface with subsequent experiments. Revisiting the experiment presented in Figure 3C with a Δ IpaH1.4 strain complemented with WT or mutated IpaH1.4 would a) show that the LPS ubiquitylation can be blocked with WT complementation, and b) validate the binding interface.

We thank the reviewer for raising this point.

We followed the reviewer's advice to complement the Δ IpaH1.4 strain. New data in the new **Extended Data Figure 4A** demonstrate that complementation with IpaH1.4 reverses the phenotype (i.e. leads to degradation of RNF213).

To address the reviewer's request regarding the IpaH1.4-RNF213 interface, we added results from two new experiments, in which we mutated each of the binding partners independently (**Extended Data Figure 8**):

Firstly, we mutated key residues in the RNF213 RING domain predicted to interact with the IpaH1.4 LRR as seen in the cryoEM structure. We then used GST-IpaH1.4 to pull down wild type and mutant RNF213 from cell lysates. We found mutations that weaken (D4013A, W4024E), and others (L4036E) that completely abolish the binding, as expected based on the cryoEM model (**Extended Data Figure 8A**).

Secondly, we mutated key residues in the IpaH1.4 LRR predicted by the cryoEM structure to interact with RNF213. We expressed those mutant IpaH1.4 proteins fused to GST in *E. coli* and tested them for binding to wild type RNF213 in pull-down experiments (**Extended Data Figure 8B**). Again, we found some mutations (R157A) that weaken, and others (K100A, F120E, R215A, V217E, F238D) that abolish the binding between IpaH1.4 and RNF213.

Taken together, new data added in response to the reviewer's comment have identified residues K100, F120, R157, R215, V217, F238 in IpaH1.4 and residue D4013, W4024, L4036 in RNF213 to mediate complex formation, thus verifying the binding surface in both proteins.

Minor concerns:

1. It is unclear from figures, figure legends, text, and methods how LPS was immunostained for western blotting.

We thank the reviewer for raising this point. In all Western blots, where we probed for ubiquitylated LPS, we first extracted the LPS from bacterial fractions isolated from infected cells through heat clearance. We then stained for conjugated ubiquitin with the FK2 anti-ubiquitin antibody (i.e. we did not stain for LPS itself). The method for LPS extraction is detailed in the Bacterial Infections section in the Methods. We have validated this method in our previous work (see Otten et al, Nature, 2021, Figure 1b-c). To further address the reviewer's comment, we have edited the figure captions in this manuscript to make this point clearer.

2. The authors propose that IpaH ligases induce proteasomal degradation of their targets by catalyzing K48 ubiquitination. Could the authors show this for RNF213? Perhaps by performing a western blot on the in vitro reaction with a K48-specific antibody?

We thank the reviewer for this suggestion. We followed the reviewer's advice and performed a Western blot with an antibody specific for K48-linked ubiquitin chains on an in vitro reaction containing IpaH1.4 or IpaH2.5 as E3 ubiquitin ligases and catalytically dead RNF213 as substrate. Results are shown in the new **Extended Data Figure 2B**. As expected, the ubiquitin chains deposited on RNF213 by IpaH1.4/2.5 are K48-linked.

3. In the introduction, the authors are missing a reference to Hansel et al., Cell 2021 for the IpaH7.8-mediated degradation of gasdermin.

We apologize for this omission, and we have now included a reference to Hansen et al, Cell, 2021, as suggested.

4. In the introduction, the authors state that "in all known cases, the IpaH ligase targets important anti-bacterial host proteins for proteasomal degradation through conjugation of K48-linked ubiquitin chains", yet the IpaH9.8 study on NEMO degradation suggested K27-linked chains.

We apologize for the over-generalization. We have modified the text accordingly: "In all known cases, the IpaH ligase targets important anti-bacterial host proteins for proteasomal degradation through conjugation of polyubiquitin chains."

5. In Figure 2C, would it be possible to also blot for RNF213 as the substrate? The lower anti-FLAG blot should be labeled with MW markers. It should also be specified which form of Ubch5 is used (the methods state Ubch5c).

We thank the reviewer for this suggestion. We labelled **Figure 2C** as advised.

We also repeated the experiment and blotted for RNF213, as suggested by the reviewer, and, additionally, for K48-linked ubiquitin chains (new **Extended Data Figure 2B**). Taken together, these results strengthen the conclusion that RNF213 is a substrate for ubiquitylation by IpaH1.4 and 2.5 in vitro, and that the attached ubiquitin chains are of the expected K48 linkage type.

6. What statistical test was used for Figure 3A?

We apologize for the omission. An unpaired t-test was performed. The symbol (*) in the chart indicates statistical significance and corresponds to a p-value of 0.018. This is now stated in the figure caption.

7. What is the difference between Figure 3B and Extended Data Figure 5?

Extended Data Figure 5 (now **Extended Data Figure 5B**) displays the individual color channels corresponding to the composite color micrographs shown in **Figure 3B**. We have edited the figure legends for clarity.

8. Given that the images presented in Figure 3B and from HeLa cells and the quantified data shown in the bar plots are from MEFs, the authors should consider splitting them into separate data panels.

We thank the reviewer for this suggestion and we have now split **Figure 3B** into two panels: B (micrographs from HeLa cells) and C (quantification from MEF cells).

9. At the bottom of page 9, I believe the authors may want to also cite Extended Data Figure S3D alongside Figure 4G.

Indeed, **Extended Data Figure 3D** shows the raw data quantified in **Figure 4G**, and we now refer to them together, as suggested.

10. In the Discussion, the sentence “The E3 ligase activity of RNF213 and IpaH-family effectors are carried” may be missing the word “out”.

We thank the reviewer for spotting our mistake. We have reworded this sentence to make it clearer: “The E3 ligase activity of RNF213 and IpaH-family effectors are mediated by the RZ and the NEL domain, respectively.”

11. In the Discussion section, the authors should consider citing Pruneda et al., Mol Cell 2012 as part of the discussion on the Arg/Lys ‘linchpin’.

We agree that this reference is required in relation to the Arg/Lys linchpin, and we have now added it to that sentence.

12. In the first sentence under the Bacterial Infections section of the Methods, there is a typo following 3.5.

We thank the reviewer for spotting this typo, and we have now corrected it.

13. In the Methods section, details on the buffer systems used for the Resource Q column were not provided.

We apologize for this omission. Details of the buffer used for anion exchange (20 mM Tris-HCl, pH 8.5, 10 – 500 mM NaCl, 4 mM DTT) were added to the Methods section.

Reviewer #2:

Remarks to the Author:

How *Shigella* may prevent LPS ubiquitination has been of great interest since the transformative discovery of LPS ubiquitination using *Salmonella* (Nature 2021). In this report, the authors discover that cytosol-dwelling *Shigella* avoids cell-autonomous immunity by secreting the bacterial effector IpaH1.4, which ubiquitinates the E3 ligase RNF213 for proteasomal degradation. Authors resolve the structure of RNF213 bound to IpaH1.4/2.5, providing high level of detail. Cellular and biochemical methods are performed (infection assays, pull-downs, protein purifications, in vitro ubiquitination, cryoEM), providing in depth mechanistic and meaningful insights on RNF213 biology. Although the role of RNF213 in *Shigella* infection control is not yet clear, this manuscript will excite researchers in the field of *Shigella* and cell-autonomous immunity. Many future research avenues emerge from this high-quality work, and I offer comments for the authors to consider for this report or for future work.

We thank the reviewer for their encouraging comments about our work and its perceived impact on understanding of *Shigella* biology and cell-autonomous immunity. We address all suggestions below.

1. To investigate degradation of RNF213, authors present a FACS based assay where they measure degradation of GFP-RNF213 in WT *Shigella* and mutants (Δ ipaH1.4, Δ ipaH2.5). It would be complementary if authors show this effect is also produced with the endogenous RNF213 measured by western blot in sorted infected cells.

We thank the reviewer for this suggestion. We have performed the suggested experiment and included the results in the new panel **Extended Data Figure 4C**. In agreement with our previous results based on GFP-tagged RNF213, we now show by Western blot that endogenous RNF213 is depleted from cells infected with *Shigella* WT and Δ ipaH2.5, but not Δ ipaH1.4. In addition, we repeated the flow cytometry-based experiment on IpaH-mediated degradation of RNF213 but we now analyze **endogenous** RNF213. We detect specific degradation of endogenous RNF213 in cells infected with *Shigella* WT and Δ ipaH2.5, but not Δ ipaH1.4 or Δ ipaH1.4/2.5 (**Extended Figure 5A**), thus confirming our original observation with GFP-tagged RNF213 in **Figure 3A**.

2. While WT *Shigella* and Δ ipaH1.4 replicate similarly during infection (Fig 3D), it seems that both strains do worse in the absence of RNF213? Could the authors explain this effect or offer any hypothesis? Also, data plotted coming from N=1 biological replicate?

We thank the reviewer for the careful inspection of the data. No differences between conditions are observed at 3 hours post infection, consistent with our interpretation of IpaH deficiency not affecting bacterial proliferation. Only at 5 hours post infection, when bacterial counts plateau (or even drop) do slight differences appear. We consider these changes within the experimental variability of the system.

3. IpaH1.4, but not IpaH2.5, is required for RNF213 degradation by *Shigella*. These results are consistent with previous work studying LUBAC. The authors suggest a downstream mechanism could be lack of IpaH2.5 secretion during infection. Considering that expression of IpaH2.5 in HeLa cells is sufficient to prevent ubiquitination of *Salmonella* LPS (Fig 2B), these cells can be used to test this hypothesis.

Our data show that over-expression of IpaH1.4 or IpaH2.5 in the host cells is sufficient to degrade RNF213 (**Figure 2B**), thereby preventing RNF213-mediated ubiquitylation of LPS (**Figure 2B**). In contrast, in the context of *Shigella* infection IpaH1.4, but not IpaH2.5, mediates RNF213 degradation (**Figure 3A**, new **Extended Data Figure 4B–C**), thereby preventing LPS ubiquitylation (**Figure 3B–D**). We initially hypothesized that failure to express IpaH2.5 might explain lack of RNF213 degradation. However, experiments revealed that both IpaH1.4 and IpaH2.5 are expressed by *Shigella* at the time of infection (**Extended Figure 4**). We therefore hypothesize that IpaH2.5 protein levels in the cytosol of infected cells must be too low to affect RNF213, for example due to lack of secretion. Regarding the reviewer's suggestion, we think there must be a misunderstanding as forced expression of IpaH2.5 in HeLa cells does not enable the testing of our hypothesis regarding insufficient IpaH2.5 protein levels in *Shigella*-infected cells.

4. Does RNF213 also ubiquitinate lipid A in *Shigella* LPS (similarly to *Salmonella*)? To test this, authors could construct a double mutant delta rfaL delta ipaH1.4.

We thank the reviewer for this question. We are also very interested in the question where exactly ubiquitin becomes attached to LPS by RNF213. However, the proposed experiment will not reveal whether *Shigella* lipid A is the ubiquitin attachment site as LPS from $\Delta rfaL$ strains merely lacks O-antigen and hence still contains many potential ubiquitylation sites in the LPS core other than lipid A. The ideal mutant for the suggested experiment, which would produce only lipid A without any core sugars attached, is unfortunately not viable and the proposed experiment thus not feasible.

Please note that our previous finding that RNF213 ubiquitylates lipid A, reported in (Otten et al, Nature, 2021), relates to purified RNF213 acting on **purified** lipid A. So, the exact ubiquitylation site on LPS **associated with bacteria** remains to be determined, both in the context of *Salmonella* and *Shigella*, or indeed any other Gram-negative bacterium.

5. Perform appropriate statistical tests for all microscopy quantifications (eg Fig 1C, Fig 3B).

We thank the reviewer for pointing out the omission. We added statistical tests to all relevant figures.

7. The authors have previously shown that, at least in the case of cytosolic *Salmonella*, LPS ubiquitination by RNF213 is required for recruitment of LUBAC to the pathogen and further M1- linked ubiquitin chain formation. This has not been demonstrated in the case of *Shigella*. Why no LUBAC testing +/- RNF213 +/- IpaH1.4 in this study?

The role of LUBAC in ubiquitylating bacteria is of great interest to us, particularly regarding the attachment site of LUBAC-generated M1-linked ubiquitin chains. However, since the topic is complex, we consider it outside the immediate scope of this manuscript, which focusses on the effects of IpaH1.4 on RNF213.

Minor points

1. The reference for the anti-LPS antibody used in Fig 1A is missing from the methods section.

We thank the reviewer for pointing this out and we apologize for this omission. We have now added the references for the anti-LPS antibodies to the Methods.

2. Provide more detail on how the fold spread of *S. flexneri* was calculated in Fig 3E.

We apologise for our oversight. The section has now been updated as follows: “...Cells were infected with a relatively low MOI of *S. flexneri* in order to achieve 2-5% infected cells at 1 h post-infection as assessed by flow cytometry. This would presumably permit each individual infected cell to spread *S. flexneri* to neighboring uninfected cells. Infection was allowed to proceed for 5 h at which point the % infected cells was quantified and the fold spread was calculated as (% infected at 5 h)/(% infected at 1 h).”

3. Fig 1A. % ubiquitination is not so black and white for Shigella? There is an extensive literature on WT *S. flexneri* associations with ubiquitin. Consistent with this, quantifications in Fig 3B is more in line with previous publications on Shigella-ubiquitin interactions.

Ubiquitin coats for Fig1c were quantified recently, while data for Fig3B are much older. Staining was therefore carried out with different batches of both the primary anti-ubiquitin antibody as well as the secondary fluorescently labelled antiserum. Batch differences may explain the observed differences. Importantly, our conclusions are unaffected by this variability as data in Fig1 and Fig3 are compared against internal controls carried out in the same experiments.

4. Previous literature has suggested that *S. flexneri* IcsB is important to avoid autophagosome recruitment. Can authors include previous literature more directly to help contextualise their results with the rest of the field. For example, how would authors explain *S. flexneri* delta icsB being recruited to autophagy in the presence of IpaH1.4?

IcsB has been proposed to inhibit specific autophagy pathways that seem distinct from the RNF213-induced ubiquitylation pathway, in particular the ability of IcsB to prevent binding of ATG5 to IcsA (VirG) on the bacterial surface (Ogawa et al, 2005) and to interact with the actin cytoskeleton via Toca-1 to antagonize recruitment of NDP52 and LC3 (Baxt and Goldberg, 2014). As requested by the reviewer, we have added references to the aforementioned publications to our manuscript.

5. Extended Fig 5 is the same as Fig 3B? Perhaps offering different images/examples would be more valuable to readers.

We thank the reviewer for raising this question. **Extended Data Figure 5** (now **Extended Data Figure 5B**) displays the individual color channels corresponding to the composite color micrographs shown in **Figure 3B**. We have edited the figure legends for clarity.

6. Crespillo-Casado et al 2024 is referenced throughout the text. I do not have access to this manuscript but hope it is available prior to publication of this work (and/or overlap is avoided).

We believe we uploaded a copy of Crespillo-Casado et al. when initially submitting this work. The manuscript by Crespillo-Casado has been accepted in the meantime and its reference has been updated.

Reviewer #3:

Remarks to the Author:

The manuscript reports the identification of IpaH1.4 as a Shigella E3 ligase effector protein that degrades the host E3 ligase RNF213. The lab recently showed that RNF213 is an important antimicrobial factor that targets LPS on Salmonella for ubiquitination, thereby signalling its degradation by xenophagy. Initial assays reveal that Shigella is not LPS ubiquitinated and identify IpaH1.4 and 2.5 as candidate RNF213 antagonists. Both bind and ubiquitylate RNF213, but only Shigella depleted of IpaH1.4 impairs GFP-RNF213 degradation. Finally, a cryoEM structure of a complex of the substrate targeting LRR region of IpaH bound to full-length RNF213 RING is solved. The LRR is contacting the RNF213 RING domain, as found with IpaH1.4 and the RING domain from HOIP. Overall, the paper is interesting but some aspects are underdeveloped and the catalytic E3 module from IpaH1.4 is not resolved. Moreover, the cellular proliferation of Shigella depleted of IpaH1.4 is unaffected. This raises an important question about the relevance of RNF213 recognition and degradation. For these reasons, and those below, it is difficult to fully support publication in NSMB.

We thank the reviewer for their concise summary of our work. We are pleased to learn that the reviewer considers our paper interesting and suggests avenues for improvement. We have addressed their concerns below and believe that the reviewer-inspired experiments have further improved our manuscript.

Major points

1. RNF213 has already been solved so the novel structural insight is the inverted configuration of the RING domain compared to HOIP RING. As it can bind RING domains in two orientations, does this imply it is promiscuous? One would expect the identification of key residues that upon mutation impair and/or selectively impair binding. These should then be tested in cells for stabilization.

We thank the reviewer for raising this point. To address the reviewer's request, we added results from two new experiments designed to validate the IpaH1.4–RNF213 interface, in which we mutated each of the binding partners independently (**Extended Data Figure 8**):

Firstly, we mutated key residues in the RNF213 RING domain predicted to interact with the IpaH1.4 LRR as seen in the cryoEM structure. We then used GST-IpaH1.4 to pull down wild type and mutant RNF213 from cell lysates. We found mutations that weaken (D4013A, W4024E), and others (L4036E) that completely abolish the binding, as expected based on the cryoEM model (**Extended Data Figure 8A**).

Secondly, we mutated key residues in the IpaH1.4 LRR predicted by the cryoEM structure to interact with RNF213. We expressed those mutant IpaH1.4 proteins fused to GST in E. coli and tested them for binding to wild type RNF213 in pull-down experiments (**Extended Data Figure 8B**). Again, we found some mutations (R157A) that weaken, and others (K100A, F120E, R215A, V217E, F238D) that abolish the binding between IpaH1.4 and RNF213.

Taken together, new data added in response to the reviewer's comment have identified residues K100, F120, R157, R215, V217, F238 in IpaH1.4 and residue D4013, W4024, L4036 in RNF213 as essential for complex formation, thus verifying the binding surface in both proteins.

2. The authors suggest other RING E3s might be targeted, which seems likely. An appreciation of the number would strengthen the manuscript. Insight could be generated with an affinity purification MS experiment using the LRR region as bait.

We thank the reviewer for the suggestion to use a mass spectrometry-based screen to investigate whether other RING-containing E3 ligases may be targeted by IpaH1.4/2.5. We have performed the suggested experiment, as described in the new section 'Mass spectrometry' in the Methods. We identified six further RING-containing E3 ligases that show preferential binding to IpaH1.4/2.5 in comparison to IpaH9.8 and other negative controls. These results are presented in a new paragraph at the end of the Results section and are shown in the new **Supplementary Table 3** and **Extended Data Figure 9**.

Computational structure predictions confirm that the IpaH1.4/2.5 LRR domain targets the E2-binding interface of the RING domain in each of the six additional interaction partners. Except for TRAF2 (de Jong et al, 2016), none of these hits has been previously described as a target of IpaH1.4/2.5, but they all are immune-related ligases, and we are interested in characterizing them in future investigations. Interestingly, we also find known interaction partners of these E3 ligases in our pulldowns (i.e. DTX3L along with PARP9; TRAF2 along with TANK and TBK1), suggesting IpaH1.4/2.5 interact with their assembled complexes in cells. We believe our finding that IpaH1.4/2.5 can target a spectrum of immune-related RING-containing E3 ligases via a common interface (the E2 binding side of the RING) will make our manuscript conceptually even more interesting.

3. IpaH1.4 can target RNF213 and HOIP. In Figure 3C what happens if HOIP is knocked out?

We thank the reviewer for this question. We are also very interested in the role of LUBAC in the ubiquitylation of cytosolic bacteria. For example, it remains unclear whether LUBAC can extend M1-linked ubiquitin chains on LPS initially ubiquitylated by RNF213, or whether the M1-linked ubiquitin chains are deposited onto some other substrate(s). We are currently pursuing this question and hope to be able to report our findings soon in another manuscript.

4. It is proposed that IpaH2.5 depletion has no effect because it isn't expressed as highly as 1.4. The authors should compare expression levels in infected cells to confirm if this is the case

We have already demonstrated that both IpaH1.4 and IpaH2.5 are transcribed by *Shigella* at the time of infection of target cells, as measured by quantitative PCR (**Extended Figure 4**). We therefore propose that IpaH2.5 is either **not translated** or **not**

secreted by bacteria. We do not currently possess the necessary tools to compare protein levels of IpaH1.4 and IpaH2.5 in infected cells, since antibodies to these proteins are not available.

We think that alternative approaches, such as generating epitope-tagged knock-in bacteria, could help to address this question. We think these experiments are beyond the scope of this manuscript. However, we hope to be able to report on this in future studies.

5. In Figure 3B why was LPS immunostaining not carried out as with 1A?

In **Figures 1A** and **3B** we use alternative methods to visualize bacteria, but the experimental results are not affected by the choice of method. In **Figure 1A** we used wild-type bacteria that are **not** fluorescent, hence the need to stain them with antibodies, and we chose to do so with an LPS antibody. In **Figure 3B** we used strains of *Shigella flexneri* that encode the fluorescent protein mRuby on a plasmid. This enables us to visualize them directly without staining. The combination of these two different approaches was required for the experiment in **Figure 1C**, where we co-infected cells with *Shigella flexneri* and *Salmonella* Typhimurium. In that case, we used LPS immunostaining to visualize **all** bacteria, and a fluorescent protein encoded by *Shigella flexneri* to discriminate them. Hence, we chose to show bacteria stained with LPS antibody in **Figure 1A** as a validation of the approach in **Figure 1C**. In all later figures we show bacteria expressing mRuby, which was also the marker used for the related FACS experiments. This is why we felt compelled to showcase with micrographs both methods used to label bacteria in this study.

6. There is no mxiE KO data in Figure 3B. Perhaps the labels are incorrect?

We thank the reviewer for raising this point. In the original submission, **Figure 3B** contained a combination of micrographs and bar graphs, which was not very clear. We have now split **Figure 3B** into two panels: **3B** (micrographs, upper) and **3C** (bar graphs, lower). The data for the mxiE knockout strain is shown in the bar graphs only. We now refer to it in the text as **Figure 3C** instead.

7. Data show that overexpressed GFP-RNF213 is degraded by Shigella. Can the authors also show that endogenous protein is degraded in infected cells?

We thank the reviewer for this question. We have addressed this point experimentally through two complementary approaches, namely (i) FACS of *S. flexneri*-infected cells and subsequent assessment of endogenous RNF213 protein levels by Western blot (**Extended Data Figure 4C**) and (ii) measurement of **endogenous** RNF213 levels in infected cells by flow cytometry (**Extended Data Figure 4B**). The new results on endogenous RNF213 agree with the approach that monitored over-expressed GFP-RNF213: both *S. flexneri* WT and Δ IpaH2.5 degrade RNF213, whereas *S. flexneri* Δ IpaH1.4 does not.

Minor points

1. In Figure 1A colocalisation is restricted to specific subregions. Is there an explanation for this?

This is a very interesting observation that we discuss in detail in Crespillo-Casado et al, 2024. Briefly, we demonstrate that RNF213 coats result from a **rare** initiation event (at a single point on a pathogen) followed by a **rapid** recruitment of further RNF213 molecules to the initiation site and spreading of the RNF213/ubiquitin coat from the initiation point via a feed-forward regulatory mechanism. We believe the interplay of slow initiation and rapid progression of coat formation results in these spatially localized coats.

2. In the discussion the argument that, because IpaH targets the RING domain implies it must have an important function, is difficult to comprehend. Isn't the RING merely serving as a recruitment site to initiate degradation? If so, the site is irrelevant and this doesn't offer any insight into the normal role of the RING domain. Could the key role of IpaH1.4 in the context of RNF213 be to block the E3 activity of the RING domain by occluding E2 binding?

We thank the reviewer for feedback on the wording of this section and we have attempted to improve it for clarity.

What we wanted to emphasize is that the RNF213 RING domain has **no** known function to date because the E3 ligase activity of RNF213 depends entirely on its RZ finger, not its RING domain. For example, **Extended Data Figure 2A** shows that RNF213 retains its autoubiquitylating activity even upon complete deletion of its RING domain. In contrast, RNF213 completely loses autoubiquitylating activity when the RZ-finger domain is mutated. We previously reported and discussed this scenario in more detail in Otten et al, Nature, 2021. However, and as pointed out by the reviewer, in the context of IpaH1.4 and RNF213, the RING domain merely serves as a recruitment site to initiate degradation of RNF213 by IpaH1.4. Our experiments with catalytically dead variants of IpaH (**Figure 2B**) show that steric hindrance of the RNF213 RING domain by inactive IpaH1.4/2.5 is insufficient to diminish LPS ubiquitylation activity. This is consistent with our previous finding (Otten et al, 2021) that LPS ubiquitylation activity does not depend on the RING domain.

Thus, we found it interesting that despite the evolutionary pressure of IpaH1.4 on RNF213, the RNF213 RING domain has not evolved away to avoid IpaH1.4 binding, which suggests that the RING domain may indeed serve an important function (perhaps, but not limited to, E3 activity under some hitherto unknown specific conditions). Interestingly, the importance of the RNF213 RING domain is also highlighted by the clustering of Moyamoya disease-related mutations in this domain. Investigation of the RNF213 RING domain function in cells remains one of our future research interests.

3. Figure legends would benefit from more information.

We thank the reviewer for this suggestion, and we have expanded the figure legends for clarity.

Reviewers' Comments:

Reviewer #1 (Remarks to the Author):

The authors have presented a nicely revised manuscript. Reviewer 3 raises a concern over the lack of a phenotype in either bacterial proliferation or spread following deletion of IpaH1.4. I am less concerned about this observation because, as the authors nicely explain, there are likely other effectors acting redundantly downstream of RNF213 to evade restriction.

We thank the reviewer for reading our revised manuscript. We are happy to hear that they are satisfied with the revisions, and we agree with Reviewer 1 that the lack of a phenotype in bacterial proliferation upon deletion of IpaH1.4 is unsurprising and most likely due to redundancy amongst bacterial effectors.

Both Reviewer 3 and I asked for additional validation of the LRR-RNF213 interface resolved in the cryoEM structure. In response, the authors have added a mutational analysis of both protein surfaces and identified residues that abrogate interactions observed by in vitro pulldowns. These data, presented in Extended Data Figure 8, could be explained better in the legend, as it isn't entirely clear which protein is being used as bait (though I believe it is GST-IpaH1.4 in both cases).

Following the reviewer's suggestion, we expanded the legend of Extended Data Figure 8 for clarity. It is indeed the case that GST-IpaH1.4 (wild-type or the indicated mutants) was used as bait. We also updated the corresponding Methods section (the second paragraph under the 'Pull down assays' subheading), and we now emphasize that this section relates to Extended Data Figure 8.

Furthermore, it isn't clear why the loading of GST-IpaH1.4 is so different across the structural mutants. Were some of the mutants less stable?

*All GST-IpaH1.4 mutants were expressed in E. coli cultures under identical conditions, and equal volumes of bacterial lysates were loaded onto the GST resin for the pulldown. Variability in the Coomassie-stained gel therefore indicates that IpaH protein levels are variable across the panel of mutants, for example due to protein stability as suggested by the reviewer. To address the reviewer's comment, and since we are solely interested in the ability of the mutants to bind RNF213 and not their expression levels in bacteria, we have now repeated the experiment using normalized amounts of GST-IpaH1.4 for the pulldown (**Extended Data Figure 8B**). New data confirm our original conclusion, namely that the mutations K100A, F120E, R215A, V217E, F238D in IpaH1.4 interfere with binding to RNF213, thus verifying the binding interface observed in the cryo-EM structure.*

Lastly, both Reviewer 3 and I asked for some of these structural mutations to be introduced back into the cellular assay for RNF213 stability. The authors nicely show that the RING domain of RNF213 is required for IpaH1.4-mediated degradation, but what about more targeted mutations? I understand that it may be technically challenging to test multiple mutations in both proteins, but could the authors simply test one or two IpaH1.4 mutants in the complementation experiment presented in Extended Data Figure 4?

*Following the reviewer's suggestion, we complemented Shigella Δ IpaH1.4 with wild type and mutant IpaH1.4 in an experiment similar to that in Extended Data Figure 4. We note that due to cloning difficulties and the time constraints on this second revision we used a different expression plasmid for introduction of IpaH1.4 into Shigella in Extended Data Figure 9 than we used formerly in Extended Data Figure 4 (pFPV compared to p4928). While the fluorescent reporter intensity from pFPV was greater than that from p4928, both gave similar IpaH1.4-mediated degradation of RNF213, and within each individual figure all comparisons are made using only the one respective plasmid backbone. The newly added results reveal that point mutations in IpaH1.4 that interfere with binding to RNF213 also reduce degradation of RNF213 in infected cells (**Extended Data Figure 9**).*

Outside of these points, I feel that the authors have satisfied all comments from both me and Reviewer 3, and I continue to strongly believe that this is important work worthy of publication in NSMB.

We thank the reviewer once again for their constructive comments and for their support of our work.

Reviewer #2 (Remarks to the Author):

Thank you for addressing reviews and for clearly advancing the field.

We would like to thank the reviewer again for their useful suggestions.

Reviewer #3 (Remarks to the Author):

The authors have made a good effort to address my concerns.

We thank the reviewer for reading our revised manuscript.

However, my primary issue was not discussed in the rebuttal which is that RNF213 has no antimicrobial activity towards *Shigella*, as evidenced from *Shigella* depleted of IpaH1.4 proliferating similarly to WT.

*Proliferation of *Shigella* Δ IpaH1.4 is unaffected by RNF213, a result that we discuss extensively in our manuscript (see below for details) and which is concordant with the well-known and frequently encountered phenomenon of functional redundancy amongst bacterial effectors. It should be noted that ubiquitylation of *Shigella* Δ IpaH1.4 itself is unlikely to be anti-bacterial and rather that downstream ubiquitin-binding proteins enact an anti-bacterial response. We therefore propose that bacterial interference with said downstream machinery is responsible for the lack of proliferative phenotype in *Shigella* Δ IpaH1.4.*

I remain unconvinced that the characterisation of what might be a redundant function is appropriate for the readership of NSMB.

*Redundancy is a well-established principle in bacterial pathogenesis. Since pathogens often translocate multiple effector proteins into hosts, eliminating individual effector genes may not result in a noticeable phenotype under experimental conditions even though individual effectors are clearly of selective advantage to the pathogen as indicated by their evolutionary conservation. For a comprehensive review see PMID 29188194 and references therein. We are deeply encouraged by support from the other two reviewers who consider our work of ‘...high interest to the ubiquitin and host-pathogen communities’ (Reviewer 1) that will ‘... excite researchers in the field of *Shigella* and cell-autonomous immunity (Reviewer 2).*

My stance is exacerbated by the abstract and summary paragraph of the introduction, which are misleading because of the narrative that RNF213 LPS ubiquitination creates "eat me signals" that must destroy all gram negative bacteria by xenophagy.

We have carefully reviewed our abstract and introduction. Neither paragraph states that 'RNF213 LPS ubiquitination creates "eat me signals" that must destroy all Gram-negative bacteria by xenophagy' (our emphasis), nor do we make such claim anywhere else in our manuscript.

To support publication, I would expect more upfront transparency regarding potential redundancy in the infection model used, and if IpaH1.4 suppresses immune responses, the benefits to the pathogen remain unclear.

We followed the reviewer's suggestion to further highlight redundancy in the pathway.

To the Results section, where we previously stated "**RNF213-mediated ubiquitylation of bacterial LPS is insufficient to restrict *Shigella* proliferation or cell-to-cell spread**" we added an explicit clarification that this observation is made in a cell-based infection model.

In the Results section we provide suggestions how *Shigella* may "...**block downstream effects of RNF213-mediated ubiquitylation. These might include factors that block the recruitment of autophagy receptors and inhibit NF- κ B activation, for example, *IcsB*, *IpaH9.8*, *OspI* and *OspZ* (Refs 23,39,40,43–45), or completely orthogonal means, such as actin-mediated motility, which may allow escape from the consequences of LPS ubiquitylation.**"

In the Discussion we further examine redundancy in the attempts of *Shigella* and *Burkholderia* to interfere with LPS ubiquitylation: "... **deletion of *IpaH1.4*, either alone or in combination with *IpaH2.5*, was insufficient for RNF213-mediated ubiquitylation to restrict *Shigella* replication in a cell-based infection model, a situation reminiscent of *TssM* in *Burkholderia thailandensis*, whose deletion also did not affect bacterial replication in response to bacterial ubiquitylation (Ref 47). Such lack of phenotype is often due to the presence of additional inhibitors, particularly in pathogens with a plethora of effectors, such as *Legionella pneumophila*, but also *Shigella*, where host pathways are simultaneously targeted by multiple effectors, for example *IpaH9.8* and *OspC3*, which inhibit GBPs and caspase-4, respectively (Ref 34,52). The specific mechanisms that ubiquitylated *S. flexneri* Δ *IpaH1.4* deploys to evade autophagy remain to be elucidated and may reveal hitherto unknown bacterial effector – host protein pairs.**"

As can be seen from the above, we believe that we have been very upfront about our results, and we discuss redundancy at length and in the context of multiple bacterial pathogens. We hope that the reviewer will agree that the work presented in this manuscript is technically sound and presented in an honest narrative, that we have addressed all their suggestions for additional experiments, and that the open questions remaining, such as the one regarding the benefits to the pathogen, provide exciting opportunities for future research, rather than shortcomings of the current work. We have added the following sentence to the discussion to convey this: "**The specific mechanisms that ubiquitylated *S. flexneri* Δ *IpaH1.4* use to evade autophagy remain to be elucidated and may reveal hitherto unknown bacterial effector – host protein pairs.**"

Reviewer #1 (Remarks to the Author):

I commend the authors for addressing all reviewer comments, yielding a study that will be highly impactful across multiple fields. I disagree with the latest comments from Reviewer 3, and feel that the authors have presented their findings in a fair and accurate manner. As the authors state, the lack of a phenotype is not uncommon in this field and only adds to the excitement of what future work will find at this host-pathogen interface.

My only remaining suggestion is for the authors to check the new figure legend for Extended Data Figure 9, which mentions a GroEl loading control in panel C. In this case, I believe the authors intended to present the actin loading control that is part of panel B.

We thank the reviewer for spotting this mistake. The loading control in panel B was indeed incorrectly labelled. Panel and legend have been edited accordingly. We would like to apologize for the mistake.